



# Evaluation of global EMEP MSC-W (rv4.34)-WRF (v3.9.1.1) model surface concentrations and wet deposition of reactive N and S with measurements

Yao Ge[1,2], Mathew R. Heal, [1], David S. Stevenson[3], Peter Wind[4], Massimo Vieno[2]

[1] School of Chemistry, University of Edinburgh, Joseph Black Building, David Brewster Road, Edinburgh, EH9 3FJ, UK

[2] UK Centre for Ecology & Hydrology, Bush Estate, Penicuik, Midlothian, EH26 0QB, UK

[3] School of GeoSciences, University of Edinburgh, Crew Building, Alexander Crum Brown Road, Edinburgh, EH9 3FF, UK

[4] The Norwegian Meteorological Institute, Henrik Mohns Plass 1, 0313, Oslo, Norway

*Correspondence to*: Yao Ge (Y.Ge-7@sms.ed.ac.uk), Mathew R. Heal (M.Heal@ed.ac.uk)

**Abstract.** Atmospheric pollution has many profound effects on human health, ecosystems, and the climate. Of concern are high concentrations and deposition of reactive nitrogen ($N_r$) species, especially of reduced N (gaseous $NH_3$, particulate $NH_4^+$). Atmospheric chemistry and transport models (ACTMs) are crucial to understanding sources and impacts of $N_r$ chemistry and its potential mitigation. Here we undertake the first evaluation of the global version of the EMEP MSC-W ACTM driven by WRF meteorology (1°×1° resolution), with a focus on surface concentrations and wet deposition of N and S species relevant
to investigation of atmospheric $N_r$ and secondary inorganic aerosol (SIA). The model-measurement comparison is conducted both spatially and temporally, covering 9 monitoring networks worldwide. Model simulations for 2010 compared use of both HTAP and ECLIPSE$_E$ (ECLIPSE annual total with EDGAR monthly profile) emissions inventories; those for 2015 used ECLIPSE$_E$ only. Simulations of primary pollutants are somewhat sensitive to the choice of inventory in places where regional differences in primary emissions between the two inventories are apparent (e.g. China), but much less so for secondary
components. For example, the difference in modelled global annual mean surface $NH_3$ concentration using the two 2010 inventories is 18% (HTAP: 0.26 µg m$^{-3}$; ECLIPSE$_E$: 0.31 µg m$^{-3}$) but only 3.5% for $NH_4^+$ (HTAP: 0.316 µg m$^{-3}$; ECLIPSE$_E$: 0.305 µg m$^{-3}$). Comparisons of 2010 and 2015 surface concentrations between model and measurement demonstrate that the model captures well the overall spatial and seasonal variations of the major inorganic pollutants $NH_3$, $NO_2$, $SO_2$, $HNO_3$, $NH_4^+$, $NO_3^-$, $SO_4^{2-}$ , and their wet deposition in East Asia, Southeast Asia, Europe and North America. The model shows better
correlations with annual average measurements for networks in Southeast Asia (Mean $R$ for 7 species: $\overline{R_7}$ = 0.73), Europe ($\overline{R_7}$ = 0.67) and North America ($\overline{R_7}$ = 0.63) than in East Asia ($\overline{R_5}$ = 0.35) (data for 2015), which suggests potential issues with the measurements in the latter network. Temporally, both model and measurement agree on higher $NH_3$ concentrations in spring and summer, and lower concentrations in winter. The model slightly underestimates annual total precipitation measurements (by 13-34%) but agrees well with the spatial variations in precipitation in all four world regions (0.65-0.78 $R$ range). High
correlations between measured and modelled $NH_4^+$ precipitation concentrations are also observed in all regions except East Asia. For annual total wet deposition of reduced N, the greatest consistency is in North America ($R$ = 0.75), followed by Southeast Asia ($R$ = 0.68) and Europe ($R$ = 0.61). Model-measurement bias varies between species in different networks; for example, bias for $NH_4^+$ and $NO_3^-$ is most in Europe and North America and least in East and Southeast Asia. The greater uniformity in spatial correlations than in biases suggests that the major driver of model-measurement discrepancies (aside from
differing spatial representativeness and uncertainties and biases in measurements) are shortcomings in absolute emissions rather than in modelling the atmospheric processes. The comprehensive evaluations presented in this study support the application of this model framework for global analysis of current and potential future budgets and deposition of $N_r$ and SIA.





## 1 Introduction

In view of increasing growth in global anthropogenic emissions, the physical and chemical behaviour of reactive nitrogen ($N_r$) species, especially those that contain reduced N (i.e. gaseous $NH_3$ and particulate $NH_4^+$) have been explored in both experimental and modelling studies (Liu et al., 2019; Wagner et al., 2020; Ciarelli et al., 2019; Tang et al., 2021). As the predominant alkaline gas, $NH_3$ exerts significant control on the formation of ambient particles and the acidity of deposition. It readily reacts with $H_2SO_4$ and $HNO_3$ (respectively derived from emissions of $SO_2$ and $NO_x$), and the ammonium sulphate

(($NH_4$)$_2SO_4$) and nitrate ($NH_4NO_3$) particles formed in these reactions are important in Earth's radiation budget (Laskin et al., 2015) due to their capacity to act as cloud condensation nuclei and to absorb/scatter solar radiation. Crucially, the ($NH_4$)$_2SO_4$ and $NH_4NO_3$ secondary inorganic aerosols (SIA) typically constitute at least a third of the fine particulate matter ($PM_{2.5}$) surface concentration (Li et al., 2017), exposure to which causes substantial premature mortality globally (Burnett et al., 2018). For half the world's population, the $PM_{2.5}$ air pollution burden is increasing (Shaddick et al., 2020). In addition, $NH_3$ and $NH_4^+$

enter aquatic and terrestrial ecosystems through wet and dry deposition where they are powerful nutrients for many plants and microorganisms. As a result, excessive anthropogenic reduced N emissions to the atmosphere can lead to severe eutrophication and formation of hypoxic zones, with their consequent threats to ecosystem diversity (Erisman et al., 2005).

The surface concentrations and deposition fluxes of atmospheric pollutants are influenced by many spatial and temporal factors such as emissions, meteorology, long-distance transport and chemical transformations. Ambient measurements play a

vital role in assessing existing concentrations but can generally only represent the air quality in the local area and cannot immediately distinguish between the influence of local and remote sources. Speciated gas and particle-phase sampling and analysis is challenging and expensive (Tang et al., 2018b). Consequently, measurements are generally sparsely located and often not very well temporally resolved, even in regions of the world with well-developed air pollution monitoring networks (Tang et al., 2021), which again limits the interpretation of atmospheric chemical and meteorological processes. Moreover,

different world regions have monitoring networks that are subject to different analytical and data handling protocols, potentially leading to systematic differences. Non-identical sampling duration and frequencies within these networks also add uncertainties and complexities to global comparison studies.

Compared with measurements, global and regional-scale atmospheric chemistry transport models such as EMEP MSC-W (Simpson et al., 2012), CMAQ (Byun and Schere, 2006) and WRF-Chem (Chapman et al., 2009) can provide comprehensive

simulations of air pollutant concentrations and depositions with greater spatial-temporal resolution and coverage. These models also facilitate insight into the chemical and meteorological linkages between diverse emission sources and the concentration and deposition of pollutants at locations away from initial emissions. Such models are essential when it comes to simulating the impacts of possible future policy actions. A number of global models have already been utilized to investigate sulphate, nitrate or ammonia budgets, including GISS II-prime (Adams et al., 1999), GEOS-Chem (Pye et al., 2009), LMDz-INCA

(Hauglustaine et al., 2014) and STOCHEM-CRI (Khan et al., 2020). Bian et al. (2017) presented a budget analysis of global nitrate simulations from 9 models and found wide variation in the tropospheric burdens of $HNO_3$, $NO_3^-$, $NH_3$ and $NH_4^+$ between the models. However, global simulations and evaluation of $N_r$ species in atmospheric chemistry transport models remain rare. In particular, there has been little comparison between modelled surface concentrations and wet deposition of $N_r$ species, especially $NH_3$ and $NH_4^+$, with regional ground-based measurement networks worldwide, which is the motivation for this

work.

Here, we present for the first time a detailed evaluation of the global simulation performance of the EMEP MSC-W chemical transport model coupled with the WRF numerical weather model. Our aim was to compare model output temporally and spatially with available ambient measurements from 9 monitoring networks in 4 global regions. A further aim was to examine the sensitivities of the model-measurement comparison to two different global emission inventories (HTAP v2 and



ECLIPSE). The primary focus of the comparisons was on atmospheric concentrations and wet depositions of the reactive N and SIA species. We also undertook evaluations for two meteorological years: 2010 and 2015.

## 2 Methods

### 2.1 Model description and set-up

The EMEP MSC-W atmospheric chemistry transport model has been developed by the European Monitoring and Evaluation Programme Meteorological Synthesizing Centre -West. As described by Simpson et al. (2012), and at www.emep.int, EMEP MSC-W is an open-source Eulerian grid model with implementation ranging from scientific research to policy development (Bergström et al., 2014; Mills et al., 2018; Karl et al., 2019; Ciarelli et al., 2019; Jonson et al., 2017; McFiggans et al., 2019). The model uses 21 terrain-following vertical layers, with the pressure ranging from around 1000 hPa (surface level) to 100

hPa (highest level). We use a lowest layer of ~45 m height. Output surface concentrations for major species are adjusted to be equivalent to 3 m above the surface as described in Simpson et al. (2012).

In this study, we utilize the most recent EMEP MSC-W model version rv4.34. Simpson et al. (2020) provide an overview of the changes made to the model since the version rv4.0 documented in Simpson et al. (2012). These changes include improved calculations of aerosol surface area and gas-aerosol uptake (Stadtler et al., 2018), additional land-cover classes and

improved leaf-area calculations for global BVOC emission calculation (Simpson, 2017), a new radiation scheme (Weiss and Norman, 1985) for BVOC and deposition calculations, new chemical mechanisms (Bergström, 2021), as well as changes related to sea-salt, dust and other emissions handling.

Most studies using EMEP MSC-W utilize meteorological data from the Integrated Forecast System model (IFS) of the European Centre for Medium-Range Weather Forecasts (ECMWF) (Fagerli et al., 2019; Pommier et al., 2020; Simpson et al.,

2012). Evaluations of the MSC-W model run with IFS meteorology can be found in Mills et al. (2018) (who found good agreement of modelled versus measured $O_3$ metrics across the GAW network), McFiggans et al. (2019) (who found good to reasonable agreement of organic aerosol data for European and North American networks), and Bian et al. (2017), who found reasonable agreement for inorganic S and N compounds in a multi-model study.

In contrast, the meteorology used for the EMEP MSC-W model simulations in this study was derived from the Weather

Research and Forecast model (WRF, www.wrf-model.org) version 3.9.1.1 (Skamarock, 2008) at grid resolution of 1° × 1°. The WRF model included data assimilation (Newtonian nudging) of the numerical weather prediction model meteorological reanalysis from the US National Center for Environmental Prediction (NCEP)/National Center for Atmospheric Research (NCAR) Global Forecast System (GFS) at 1° resolution, every 6 hours (Saha et al., 2010). A higher resolution UK/Europe regional version of the EMEP-WRF modelling system has been well evaluated previously against field measurements (Vieno

et al., 2010; Vieno et al., 2016; Vieno et al., 2014). However, an assessment of the global version has not yet been undertaken. Moreover, integrating WRF with the EMEP MSC-W model is still an innovative application, as most studies utilize meteorological data from the IFS model as described above.

Two global emission inventories were used in this work. The ECLIPSE (Evaluating the CLimate and Air Quality ImPacts of Short-livEd Pollutant) inventory version V6 (https://iiasa.ac.at/web/home/research/researchPrograms/air/ECLIPSEv6.html)

contains annual gridded emissions of $SO_2$, $NO_2$, $NH_3$, CO, $CH_4$, NMVOC (non-methane volatile organic compounds), primary fine particulate matter ($PM_{2.5}$) and primary coarse particulate matter ($PM_{co}$) (Klimont et al., 2017) at 0.5° × 0.5° spatial resolution. Its emission sectors include energy, industry, solvent use, transport, domestic combustion, agriculture, open burning





of agricultural waste, and waste treatment. We used ECLIPSE emission inventories for 2010 and 2015 to permit comparison between model and measurements for two self-consistent years of emissions, meteorology and measurements. The HTAP
(Task Force on Hemispheric Transport of Air Pollution) inventory version V2 (https://edgar.jrc.ec.europa.eu/dataset_htap_v2) consists of $0.1° \times 0.1°$ gridded monthly emissions of $SO_2$, $NO_2$, $NH_3$, CO, $CH_4$, NMVOC, $PM_{2.5}$, $PM_{10}$, black carbon (BC) and organic carbon (OC) for 2010 (2015 was not available at the time of this work) from 7 sectors (international and domestic air, shipping, energy, industry, transport, residential, and agriculture) and was used to investigate the sensitivity of model outputs to different global inventories. The HTAP inventory utilises nationally reported emissions together with regional
scientific inventories (e.g. from US-EPA, the MICS-Asia group, EMEP/TNO, the REAS and the EDGAR group) for those regions where national emissions are not available (Janssens-Maenhout et al., 2015; Gusev et al., 2012; West et al., 2010).

Both inventories were aggregated to $1° \times 1°$ resolution internally in the model. All inventory emission sector-layers were re-assigned to 11 Selected Nomenclature for sources of Air Pollution (SNAP) sectors: (1) combustion in energy and transformation industries, (2) non-industrial combustion plants, (3) combustion in manufacturing industry, (4) production
processes, (5) extraction and distribution of fossil fuels and geothermal energy, (6) solvent and other product use, (7) road transport, (8) other mobile sources and machinery, (9) waste treatment and disposal, (10) agriculture, (11) other sources and sinks.

In addition, monthly emission time series by sector and country derived from EDGAR (Emission Database for Global Atmospheric Research, v4.3.2 datasets) temporal emission profiles (Crippa et al., 2020)
(https://edgar.jrc.ec.europa.eu/dataset_temp_profile) were applied to the ECLIPSE annual total emissions for all pollutants. Therefore, from here on we refer to the inventory with ECLIPSE annual emissions and EDGAR monthly temporal profiles as $ECLIPSE_E$. All EDGAR emission subsectors (~33) are further divided into 11 SNAP sectors. The time-splitting factor ($T_{SNAP}$) for a given pollutant for a given country/region was computed as follows. Annual average emission of pollutant from EDGAR v4.3.2 subsector $j$, $\bar{P}_j$:

$$\bar{P}_j = \frac{\sum_{i=1}^{12} P_{ij}}{12}$$

Monthly time-splitting factor of pollutant from subsector $j$, $T_{E\_j}$:

$$T_{E\_j} = \frac{P_{ij}}{\bar{P}_j}$$

The weight of $T_{E\_j}$ in month $i$:

$$W_{ij} = \frac{P_{ij}}{\sum_{j=1}^{n} P_{ij}}$$

The time-splitting factor for the EMEP MSC-W model SNAP sector in month $i$:

$$T_{SNAP} = \frac{\sum_{j=1}^{n} T_{E\_j} \times W_{ij}}{\sum_{j=1}^{n} W_{ij}}$$

Forest and vegetation fire emissions and international shipping emissions are also included in both inventories. Emissions of dimethyl sulphide (DMS), lightning $NO_x$, soil $NO_x$ and isoprene are set as reported in Simpson et al. (2017; 2020) as are the wind-derived emissions of dust and sea salt (Simpson et al., 2012; Tsyro et al., 2011).






## 2.2 Measurement Datasets

Ambient measurement data were compiled from the 9 regional and national monitoring networks in East Asia, Southeast Asia, Europe, and North America listed in Table 1. The number of monitoring sites in each network varies with year and with species but Fig. 1 shows the monitoring sites for $NH_4^+$ in 2015 as an example. The frequency and duration (i.e. averaging) of sampling,
and the sampling and analytical methods used, including the size fraction of PM sampled, vary across the measurement networks. Some measurement locations are also deliberately sited to be close to particular industrial or agricultural sources, in which case a model grid average concentration may not reflect the measurement. Although much of this information is presented in official network reports, much useful metadata is absent from the data portals and addition of this information directly to the portals is a recommendation for improvement. In this work, only measurement data with at least 75% data
capture in the year are used to avoid bias. A full data mining of global measurement data was not undertaken here, but we believe we have captured the major networks of long-running, multi-species SIA gas and particle composition and wet deposition measurements.

The Chinese national nitrogen deposition monitoring network (NNDMN) was established in 2010 to measure inorganic N concentrations and deposition fluxes. The first database, NNDMN 1.0, which compiles monthly air concentration and
deposition data for $NH_3$, $NO_2$, $HNO_3$, $NH_4^+$, and $NO_3^-$ up to 2015 was released in May 2019 (Xu et al., 2019).

The acid deposition monitoring network in East Asia and Southeast Asia (EANET) involves 13 countries and provides annual and monthly concentration and acid deposition data for more than 10 species.

The UK Acid Gases and Aerosol Monitoring Network (AGANet, 30 sites) provides long-term national and monthly speciated measurements of acid gases ($HNO_3$, $SO_2$, HCl) and aerosol components ($NO_3^-$, $SO_4^{2-}$, $Cl^-$, $Na^+$, $Ca^{2+}$, $Mg^{2+}$) (Tang
et al., 2018b). The UK National Ammonia Monitoring Network (NAMN, 95 sites) includes both AGANet and additional sites with monthly measurements of $NH_3$ and $NH_4^+$ (Tang et al., 2018a). Both NAMN and AGANet provide monthly average concentrations.

The European Monitoring and Evaluation Programme/Chemical Co-ordinating Centre (EMEP/CCC) is a collaborative programme for measuring air pollutants across Europe (Tørseth et al., 2012). The measurement frequency varies from hourly
and daily to weekly and biweekly or intermittently such as every 6-days. It also varies between species. This makes it difficult to derive consistent annual and monthly averages comparisons between measurement and model.

The Air Data of the United States Environmental Protection Agency (EPA) provides access to annual outdoor air quality data including $SO_2$, $NO_2$, $NH_4^+$, $NO_3^-$, $SO_4^{2-}$, collected from state, local and tribal monitoring agencies across the United States. The Ammonia Monitoring Network (AMoN) and National Trends Network (NTN) are two further US networks that provide
long-term records of weekly/biweekly $NH_3$ gas concentrations and annual precipitation chemistries respectively.

In Canada, the National Air Pollution Surveillance (NAPS) program is the main source of ambient air quality data and consists of continuous and time-integrated monitoring of several species. Continuous measurements are implemented for CO, $NO_2$, NO, $NO_x$, $O_3$, $SO_2$, $PM_{2.5}$, and $PM_{10}$ at hourly resolution. The time-integrated samples collect once per 6 days for a 24-h period, encompass fine ($PM_{2.5}$) and coarse ($PM_{2.5-10}$) aerosol components (e.g., inorganic ions, metals), semi-volatile organic
compounds and VOCs.

The calculations of model-measurement comparison statistics (e.g. Pearson's correlation coefficient, mean bias, mean absolute error, etc.) are shown in the Supplementary Material.





**Table 1.** Summary of surface monitoring networks used in the model-measurement comparisons.

| Region | Network | Source |
|---|---|---|
| East & Southeast Asia | NNDMN (China) | https://www.nature.com/articles/s41597-019-0061-2 |
| | EANET | https://www.eanet.asia |
| Europe | AGANet (UK) | https://uk-air.defra.gov.uk/networks/network-info?view=ukeap |
| | NAMN (UK) | https://uk-air.defra.gov.uk/networks/network-info?view=nh3 |
| | EMEP/CCC | http://ebas.nilu.no/Default.aspx |
| North America | NAPS (Canada) | https://www.canada.ca/en/services/environment/weather/airquality.html |
| | EPA-Air Data (US) | https://www.epa.gov/outdoor-air-quality-data |
| | AMoN (US) | http://nadp.slh.wisc.edu/data/AMoN/ |
| | NTN (US) | http://nadp.slh.wisc.edu/NTN/ |

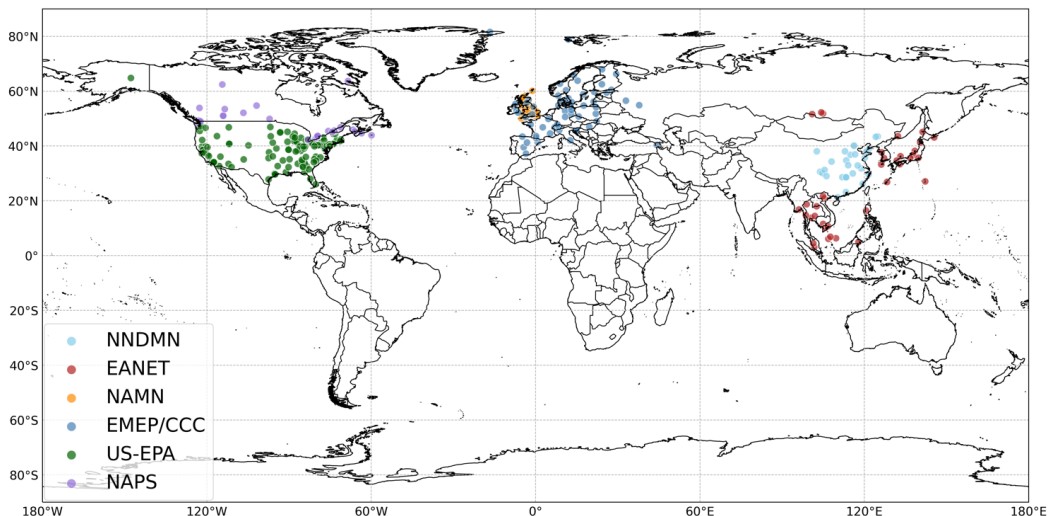

**Figure 1.** Locations of sites in the 6 networks that measured particle-phase $NH_4^+$ in 2015.


## 3 Results

### 3.1 Comparison between use of HTAP and ECLIPSE$_E$ emission inventories

#### 3.1.1 Emissions

The global map of 2010 annual $NH_3$ emissions from ECLIPSE$_E$ is shown in Fig. 2 (top). Hot spots of $NH_3$ emissions occur
across the globe in areas characterized by dense populations and intensive agricultural activities, most notably in the Indo-
Gangetic Plain in India and the North China Plain, but also in Indonesia, Europe, United States, Mexico, and Brazil. The area-
weighted average $NH_3$ emissions (over the whole global domain) in 2010 are 105 and 121 mg m$^{-2}$ for HTAP and ECLIPSE$_E$



respectively. The individual grid annual NH$_3$ emission in 2010 varies from 0.00 to 10692 mg m$^{-2}$ for the HTAP inventory and from 0.00 to 12244 mg m$^{-2}$ for the ECLIPSE$_E$ inventory. (Note that in the following sections all emissions and concentrations are expressed as mass of the species unless otherwise stated, e.g. as μgN m$^{-3}$).

Figure 2 (bottom) maps the differences in annual NH$_3$ emissions between the ECLIPSE$_E$ and HTAP inventories for 2010. Clear differences between the two emission inventories are observed in China, India, and several Southeast Asian countries, but differences in other world regions are relatively small: more than 70% of the relative differences in ECLIPSE$_E$ − HTAP emissions, the majority of which are positive, are within ± 10% of the average inventory emission for that grid. The ECLIPSE$_E$ inventory NH$_3$ emissions are larger than the HTAP inventory emissions in north and southeast parts of China, western coastal area of continental Europe, central Africa, Brazil and Argentina. The largest difference of 6496 mg m$^{-2}$, which is 73% of the inventory mean emission of 8956 mg m$^{-2}$ for that model grid, is in the east of China (Fig. 2 bottom). In contrast, HTAP reports larger NH$_3$ emissions than ECLIPSE$_E$ in areas of Southeast Asia, India, and western United States. The largest negative difference of -4281 mg m$^{-2}$ (equating to 124% of the grid mean 3452 mg m$^{-2}$) is located on the west coast of the United States. Relative NH$_3$ emission differences that are outside of ±100% of the average NH$_3$ emissions from the two inventories for that grid only account for 13% of total gridded differences, and the majority of instances where relative difference is large are for grids that have only low emissions, for which a small absolute difference equates to large relative difference.

Aside from the instances of quite localised discrepancies in the NH$_3$ emissions between the two inventories, the small median positive (7.90 mg m$^{-2}$) and negative (-12.0 mg m$^{-2}$) differences, together with the global area-weighted average difference of only 16.0 mg m$^{-2}$ (14% relative to the mean emission of the two inventories), indicate that ECLIPSE$_E$ and HTAP provide very similar annual NH$_3$ emissions in most grids over the whole global domain.

The seasonal profile of spatially averaged monthly NH$_3$ emissions of the two inventories in 2010 was also investigated for East Asia, Southeast Asia, Europe and North America separately. The detail is presented in Supplementary Material. Clear NH$_3$ emission peaks in spring and summer are observed in both inventories for all four global regions. In general, ECLIPSE$_E$ shows greater monthly variations than HTAP in East Asia, Southeast Asia, and Europe except for North America, which is strongly indicative of different monthly (or day-of-week) temporal factors applied to annual totals in different inventories.

Similar observations derive from comparisons of emissions of NO$_x$ and SO$_x$ in the two inventories (Supplementary Material Fig. S1 and S2). For example, the global area-weighted average difference in annual NO$_x$ emissions between the two inventories is only 11.0 mg m$^{-2}$ (2.9%), whilst the maximum positive and negative differences for an individual model grid (ECLIPSE$_E$ − HTAP) are 15389 mg m$^{-2}$ (162%) and −26815 mg m$^{-2}$ (−186%) respectively. These large local differences in NO$_x$ emissions are presumably due to the inclusion or exclusion of a specific point source in one emission inventory but not the other. The shipping emission profiles included in the two inventories are also slightly different. For instance, ECLIPSE$_E$ provides higher NO$_x$ emissions in the Yellow Sea, South China Sea and Bay of Bengal than HTAP (Fig. S1). Therefore, the differences between the two inventories may not have a large influence on global simulations but may have larger impact on regional modelling at higher spatial resolution.

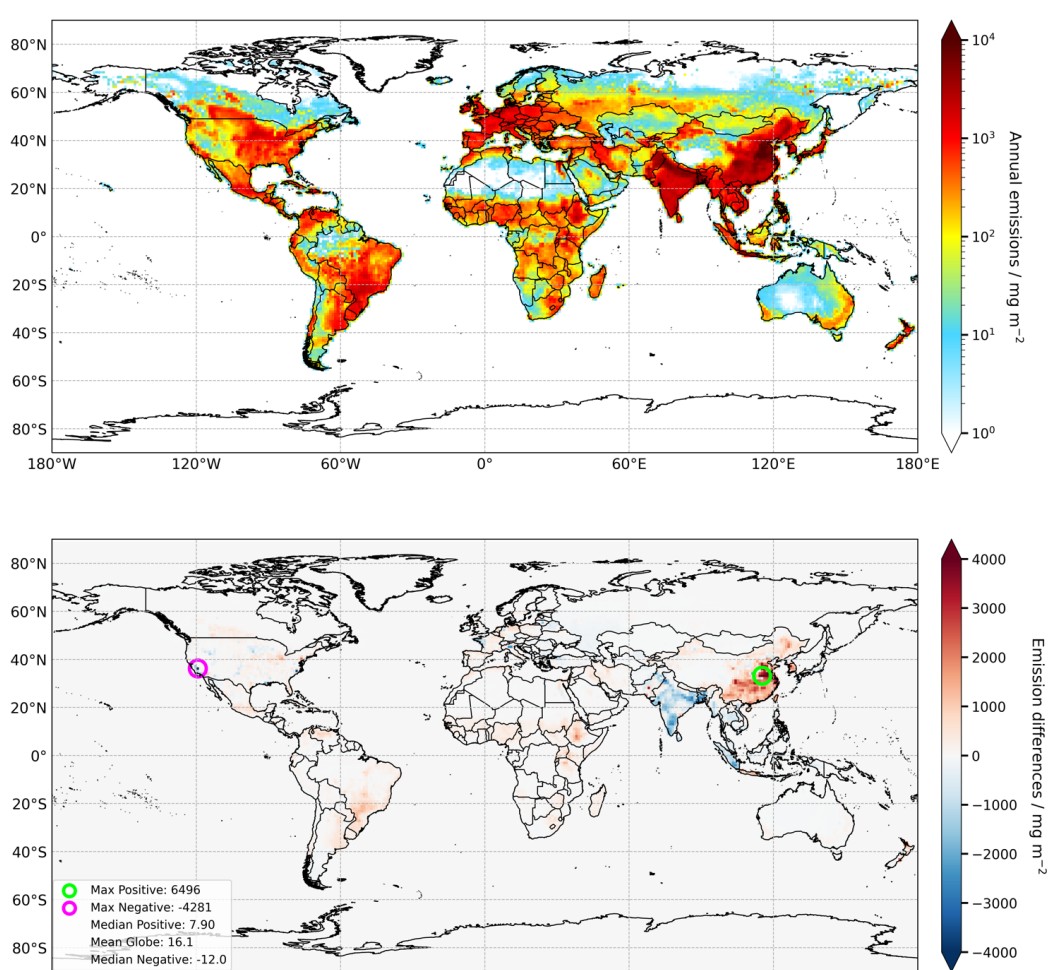

**Figure 2. Top: Global annual NH₃ emissions for 2010 from ECLIPSE_E. Bottom: the difference in 2010 annual NH₃ emissions (mg m⁻²) between ECLIPSE_E and HTAP (ECLIPSE_E − HTAP). The inset panel provides the maximum, median and mean values of both positive and negative differences across individual emission grids.**


### 3.1.2 Reduced N Concentrations

Figure 3 presents examples of the global model output: maps of the global distributions of annual mean surface concentrations and total (wet + dry) depositions of reduced N (i.e. $NH_3 + NH_4^+$) in 2010 using the ECLIPSE_E inventory. Largest reduced N

concentrations (Fig. 3 top) are located in regions of high NH₃ emissions (shown in Fig. 2): notably eastern China, northern India and Indonesia, followed by northern Italy, Germany, Midwest United States and southern Brazil. Reduced N concentrations reach ~35 µgN m⁻³ in parts of China. Annual deposition of reduced N (Fig. 3 bottom) shows clear decreasing gradients from continental regions to surrounding oceans with maxima of 5000-5200 mgN m⁻² in eastern and southern Asia and 1800-2000 mgN m⁻² in central Europe and Midwest and South United States. These regions are characterised not only by

high emissions of reduced N but also large emissions of SOₓ and NOₓ (Figs. S1 and S2), reflecting areas of greatest anthropogenic activities. Our spatial patterns of reduced N species are consistent with other global modelling studies





(Hauglustaine et al., 2014; Xu and Penner, 2012; Pringle et al., 2010). The model-measurement comparisons we carry out for this study cover the majority of these hot spot regions.

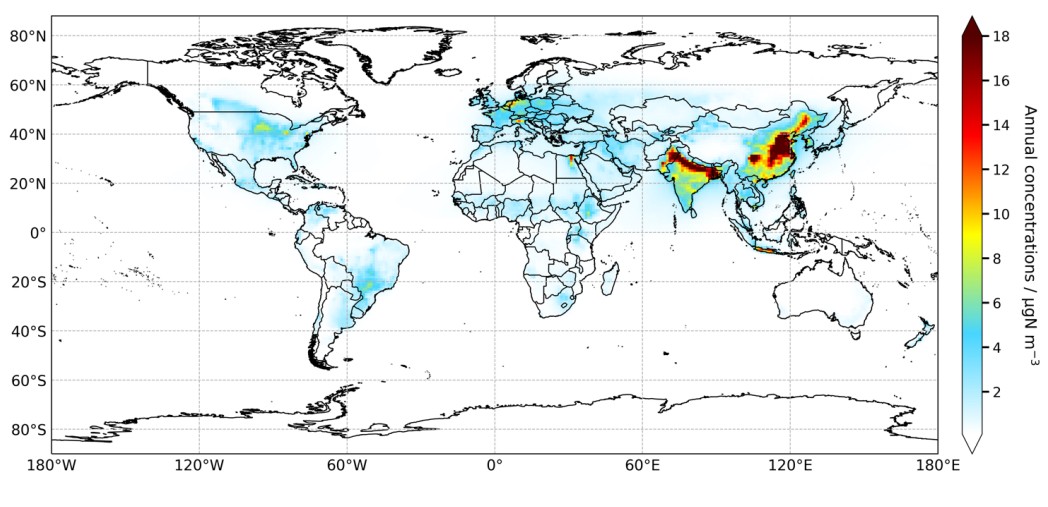

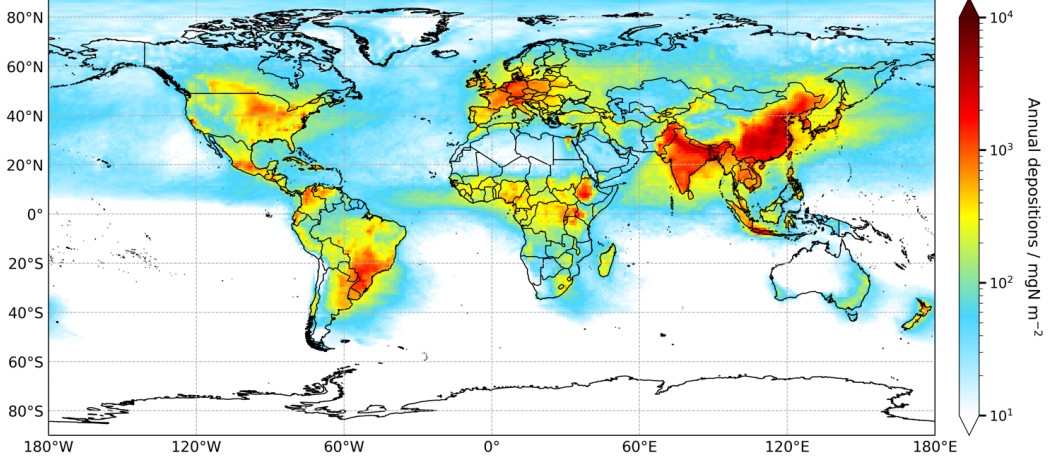


**Figure 3. (Top) Annual mean surface concentrations and (bottom) annual total (wet + dry) depositions of reduced N (NH$_3$ + NH$_4^+$) for 2010 based on the ECLIPSE$_E$ inventory.**

The influences of the two emission inventories on model simulated surface concentration differs according to consideration of primary or secondary component and varies from one region to another. Globally, the difference in modelled area-weighted annual mean surface NH$_3$ concentration using the two 2010 inventories is 18% (HTAP: 0.26 µg m$^{-3}$; ECLIPSE$_E$: 0.31 µg m$^{-3}$). The relative difference is the same when considering land-only area-weighted mean surface NH$_3$ concentration (0.83 and 0.99 µg m$^{-3}$ for HTAP and ECLIPSE$_E$ respectively). In contrast, the difference for global area-weighted mean surface NH$_4^+$ concentration is only 3.5% for NH$_4^+$ (HTAP: 0.316 µg m$^{-3}$; ECLIPSE$_E$: 0.305 µg m$^{-3}$), or 5.0% for the
land-only area-weighted NH$_4^+$ concentrations of 0.755 and 0.718 µg m$^{-3}$, respectively.

For a regional perspective, Fig. 4 and Fig. S5 respectively compare the modelled NH$_3$ and NH$_4^+$ concentrations using the two emission inventories for the grids in which there are also available measurements from the monitoring networks.





Considering all measurement locations globally, the model simulated concentrations using the two inventories are extremely well spatially correlated with each other at $R = 0.95$ for $NH_3$ and 0.98 for $NH_4^+$. The average difference in global surface $NH_3$ concentration between model simulations using ECLIPSE$_E$ and HTAP based on measurement locations is 0.34 µg m$^{-3}$, which corresponds to only 15% of the model average concentration of 2.30 µg m$^{-3}$ using the ECLISPSE$_E$ inventory or 17% of the model average concentration of 1.96 µg m$^{-3}$ using the HTAP inventory.

The model concentrations using the two emission inventories are similarly linearly correlated with measurements (Fig. 4). As discussed above, systematic differences between modelled and measured concentrations of $NH_3$ in East Asia and Southeast Asia can be attributed at least in part to local differences in $NH_3$ emissions among different inventories. The average modelled $NH_3$ concentrations in China derived from ECLIPSE$_E$ and HTAP (based on measurement locations) are 12.3 and 7.9 µg m$^{-3}$ respectively. The systematically greater modelled $NH_3$ concentrations using ECLIPSE$_E$ compared to HTAP is consistent with the ECLIPSE$_E$ inventory's larger $NH_3$ emissions over eastern and southern China (Fig. 2), where the majority of the NNDMN measurement sites are located (Fig. 1).

For measurement locations in Southeast Asia, Fig. 4 shows that modelled $NH_3$ concentrations are generally lower than their respective measured concentrations, for simulations using both emissions inventories. However, as for China, model simulations of $NH_3$ using the two inventories are spatially well correlated with each other ($R = 0.92$). The overall average modelled $NH_3$ concentration (based on grids containing EANET sites) of 1.99 µg m$^{-3}$ using the HTAP inventory is slightly greater than the average concentration of 1.50 µg m$^{-3}$ using the ECLIPSE$_E$ inventory. Using the HTAP inventory also gives a slightly larger range in simulated $NH_3$ concentrations (0.00-9.14 µg m$^{-3}$) for the grids with measurement sites than the range (0.01-6.54 µg m$^{-3}$) when using the ECLIPSE$_E$ inventory. This is again consistent with the smaller emissions for ECLIPSE$_E$ in most south-eastern Asian countries in 2010 (Fig. 2).

In North America and Europe there are similar linearities between the modelled and measured $NH_3$ concentrations when using either of the HTAP and ECLIPSE$_E$ inventories (Fig. 4). In general, both inventories produce smaller concentrations than measurements in Europe, with ECLIPSE$_E$ underestimating more, and higher concentrations than measurements in North America, with ECLIPSE$_E$ overestimating more. In other words, the ECLIPSE$_E$ inventory yields smaller $NH_3$ concentrations in Europe but higher concentrations in North America compared with the HTAP inventory. The differences in $NH_3$ emissions between the two inventories are very similar in these two regions: Fig. 2 shows that the differences in emissions are generally close to zero and that differences are both positive and negative. Therefore, it is the location of the measurement site that likely influences the model evaluation statistics. The modelled $NH_3$ concentrations in North America (based on network locations) are in the ranges 0.01-3.30 µg m$^{-3}$ and 0.04-3.64 µg m$^{-3}$ for simulations with HTAP and ECLIPSE$_E$ inventories respectively, while in Europe the equivalent modelled $NH_3$ concentration ranges are 0.00-4.36 µg m$^{-3}$ and 0.00-3.95 µg m$^{-3}$. The average $NH_3$ concentration difference (based on network locations) in North America between the two emission inventories is 0.47 µg m$^{-3}$ (ECLIPSE$_E$ – HTAP), whilst this difference in Europe is only 0.03 µg m$^{-3}$.

The impact of emission inventory differences on concentrations of secondary pollutants is much smaller than for primary pollutants since the former are influenced by multiple emissions and the timescales for their formation act to smooth out spatial differentials in primary emissions. This is illustrated by the generally better agreement between model outputs for both the HTAP and ECLIPSE$_E$ emissions inventories and the network measurements of annual $NH_4^+$ concentrations in Fig. S5 than for $NH_3$ in Fig. 4. Thus, the correlations between modelled and measured $NH_4^+$ at all network locations are 0.88 (range 0.54-0.92 for the four separate regions) and 0.90 (0.74-0.90) for simulations using the HTAP and ECLIPSE$_E$ inventories, respectively, whilst the corresponding correlation coefficients for $NH_3$ are 0.66 (0.40-0.69) and 0.68 (0.49-0.77).



The differences in $NH_4^+$ concentrations in simulations using the two emission inventories (Fig. S5) are also smaller than for $NH_3$ (Fig. 4), as shown by concentrations that are closer to 1:1 in all regions. For example, whilst modelled $NH_3$ concentrations in China derived using the $ECLIPSE_E$ inventory are on average 56% higher than those derived using the HTAP

inventory, the $NH_4^+$ concentrations are very similar. The annual average $NH_4^+$ concentrations (based on network locations) in China are 7.30 and 7.15 $\mu g\ m^{-3}$ for HTAP and $ECLIPSE_E$ respectively, which is a difference of only 2%. More detail is presented in Supplementary Material.

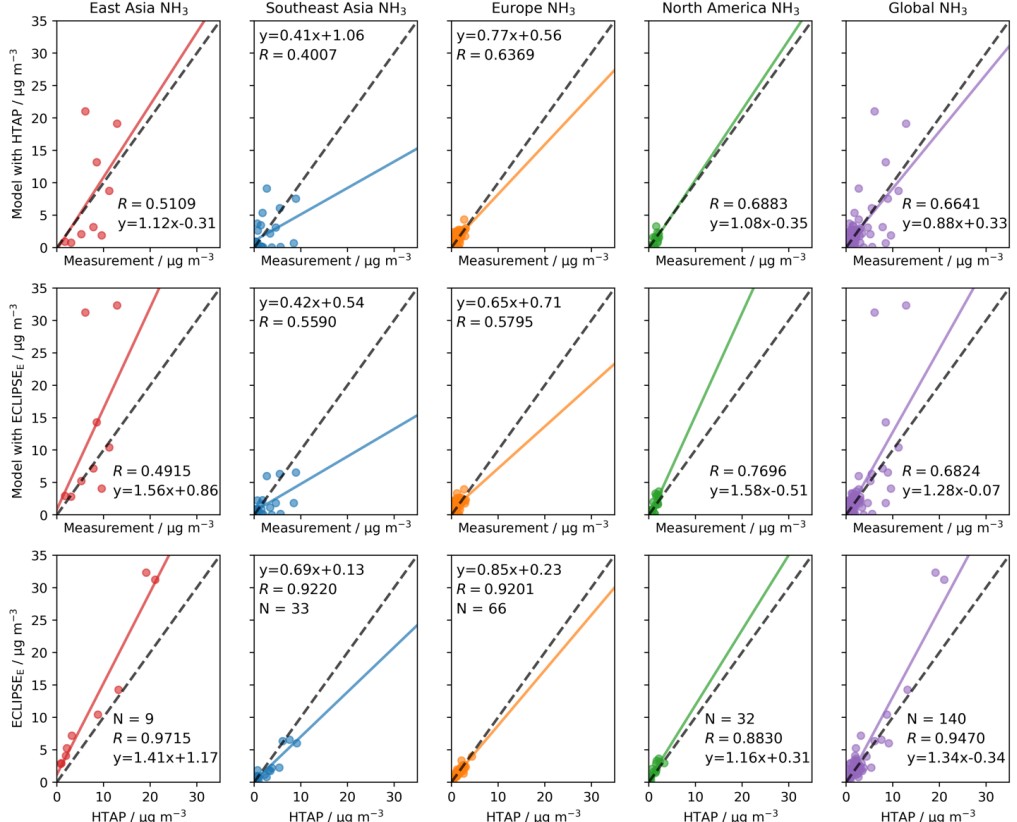

**Figure 4. Comparisons of annual average surface concentrations of $NH_3$ for 8 monitoring networks in 2010 – NNDMN from China as East Asia, EANET as Southeast Asia, NAMN and AGANet (UK) and EMEP/CCC plotted together here as Europe, and the EPA and AMoN (USA) and NAPS (Canada) plotted together here as North America – and for all networks combined ('global'). The upper row of plots is modelled versus measured using the HTAP emission inventory. The middle row is modelled versus measured using the $ECLIPSE_E$ emission inventory. The lower row is the modelled data for the two inventories plotted against each other for**
**the same set of model grids that contain measurement sites. In each plot, N is the total number of scatter points, $R$ is the Pearson correlation coefficient, the black dashed line is the 1:1 line and the coloured solid line is the trend line corresponding to the equation presented.**

In summary, whilst there are some spatial differences in annual emissions between the HTAP and $ECLIPSE_E$ inventories,
e.g. for $NH_3$ emissions in China and India, emission differences on a global scale are small. The difference in global average $NH_3$ emissions (for 2010) is 16.0 $mg\ m^{-2}$ ($ECLIPSE_E$–HTAP) which is 14% of the average of the HTAP and $ECLIPSE_E$ global mean $NH_3$ emissions of 105 and 121 $mg\ m^{-2}$, respectively. The spatial heterogeneity in the positive and negative differences in emissions worldwide indicates no global bias between them. The regional differences in emissions between the two



inventories impact differently on modelled surface concentrations of primary and secondary species. Both inventories yield
model results that show similar linear correlations with ambient $NH_3$ and $NH_4^+$ concentration measurements and similar underestimations/overestimations in different monitoring networks. The seasonality in $NH_3$ emissions of HTAP and ECLIPSE$_E$ are similar, although the latter projects greater monthly fluctuations in East Asia, Southeast Asia, and Europe, but not North America, which indicates discrepancies in temporal (monthly or day-of-week) factors applied to annual totals in different inventories.


### 3.2 Comparisons of modelled surface concentrations of N$_r$ and SIA species with measurements

Evaluations of modelled versus measured concentrations were undertaken for both 2010 and 2015. The comparisons for the two years show similar characteristics. To avoid repetition, the following section presents and discusses the comparisons for 2015, using the ECLIPSE$_E$ inventory, as more measurement data were available for 2015.

### 3.2.1 East and Southeast Asia

Figure 5 shows the spatial distribution of modelled and measured 2015 annual average $NH_3$ concentrations for regions covered by the NNDMN (China) and EANET (East Asia) networks. Scatter plots of the paired model versus measurement annual concentrations for $NH_3$, $NH_4^+$ and other gaseous and particle-phase inorganic components are shown in Fig. 6, illustrating the extent of model-measurement spatial correlations. A summary of model evaluation statistics is presented in Table 2.


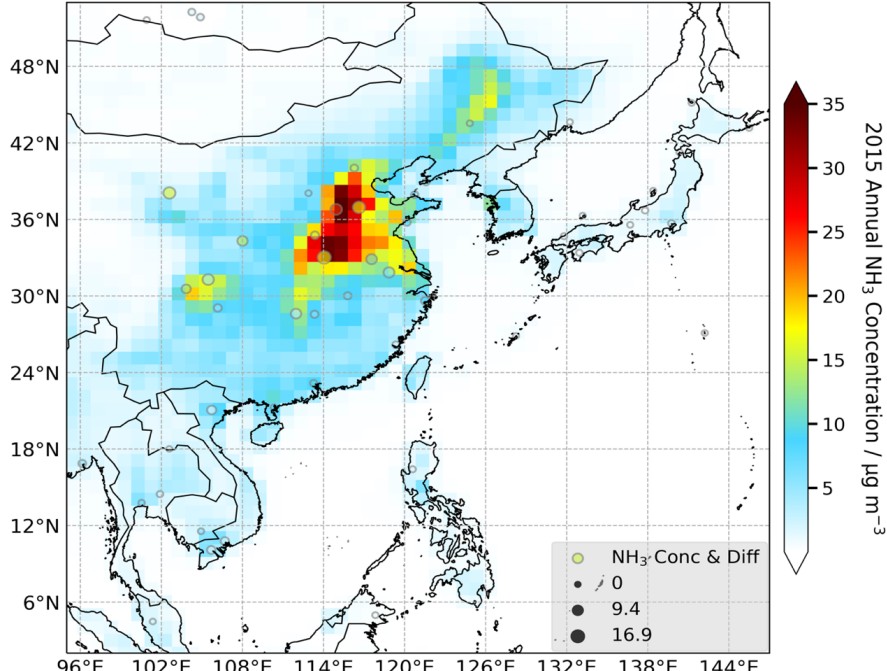

**Figure 5. Modelled and measured 2015 annual mean NH$_3$ concentrations in East and Southeast Asia. Measurements are from the China NNDMN and East Asia EANET networks. Monitoring sites are indicated by circles whose colour represents the measured concentration and whose diameter (see inset legend) represents the absolute difference between model and measurement.**






**Table 2. Summary statistics of model comparison with measurements for 2015 in East and Southeast Asia (NNDMN and EANET networks).** $N$ is the number of paired data of model and observation. $R$ is Pearson's coefficient. Fac2 is the fraction of data points within a factor of 2. Mean_O and Mean_M are annual average concentrations ($\mu g\ m^{-3}$) of observation and model respectively. NMB is normalized mean bias, NME is normalized mean error.

| NNDMN | $N$ | $R$ | Fac2 fraction | Mean_O | Mean_M | NMB | NME |
|---|---|---|---|---|---|---|---|
| $NH_3$ | 24 | 0.68 | 0.75 | 10.1 | 13.0 | 0.29 | 0.57 |
| $NO_2$ | 24 | 0.59 | 0.83 | 23.5 | 28.6 | 0.22 | 0.39 |
| $HNO_3$ | 24 | -0.18 | 0.21 | 4.90 | 1.93 | -0.61 | 0.64 |
| $NH_4^+$ | 23 | 0.42 | 0.78 | 8.10 | 8.12 | 0.00 | 0.46 |
| $NO_3^-$ | 24 | 0.26 | 0.71 | 10.0 | 13.8 | 0.38 | 0.62 |
| EANET | $N$ | $R$ | Fac2 fraction | Mean_O | Mean_M | NMB | NME |
| $NH_3$ | 27 | 0.56 | 0.52 | 1.63 | 1.92 | 0.18 | 0.69 |
| $NO_2$ | 7 | 0.84 | 0.71 | 15.6 | 25.9 | 0.67 | 0.68 |
| $HNO_3$ | 28 | 0.81 | 0.39 | 0.63 | 1.33 | 1.13 | 1.19 |
| $SO_2$ | 36 | 0.71 | 0.44 | 2.96 | 3.31 | 0.12 | 0.81 |
| $NH_4^+$ | 29 | 0.73 | 0.62 | 0.75 | 1.19 | 0.59 | 0.73 |
| $NO_3^-$ | 29 | 0.73 | 0.38 | 1.10 | 0.89 | -0.19 | 0.67 |
| $SO_4^{2-}$ | 29 | 0.74 | 0.83 | 3.03 | 2.71 | -0.11 | 0.31 |


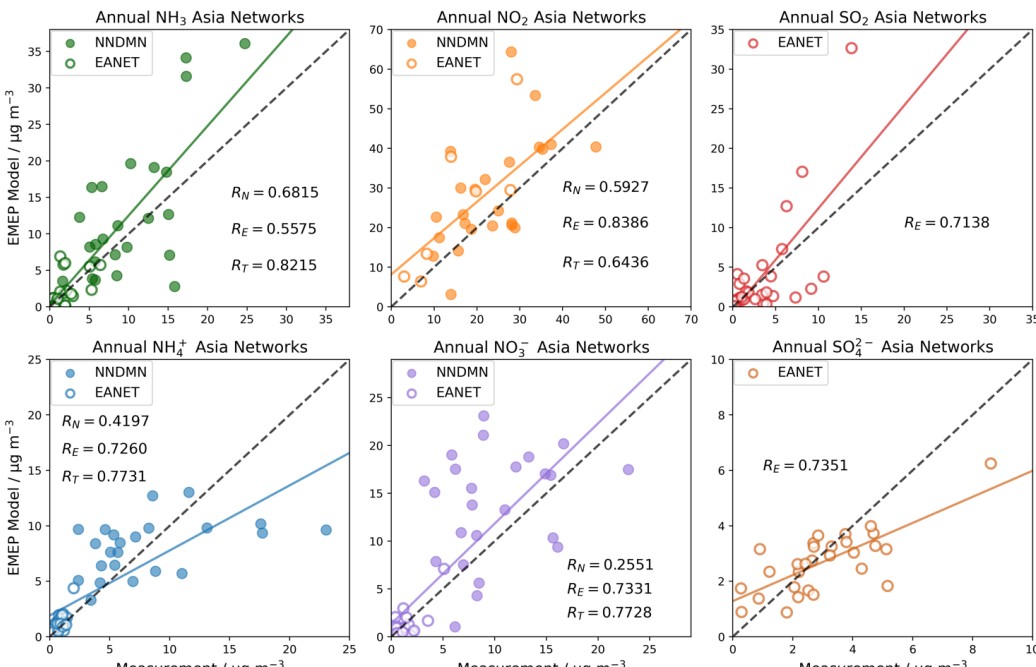

**Figure 6.** Scatter plots of 2015 annual mean modelled and measured $NH_3$, $NO_2$, $HNO_3$, $SO_2$, $NH_4^+$, $NO_3^-$, and $SO_4^{2-}$ concentrations at East and Southeast monitoring network locations. In each plot, the solid line is the least-squares regression line and the dashed black line is the 1:1 line. $R_T$ is the overall correlation coefficient between model and all measurements.






The model simulates well the overall spatial trend of annual $NH_3$ concentrations in this region, spanning a range from around 0 to >30 $\mu g\ m^{-3}$. Model and measurements consistently show highest $NH_3$ concentrations in central eastern China (typically >15 $\mu g\ m^{-3}$). The observed $NH_3$ hotspots in North China Plain, Northeast China Plain, and Sichuan Basin are consistent with them being regions of intensive agricultural activities that apply large amounts of fertilizers (Xu et al., 2015).

Most areas in other East and South-East Asia countries such as Japan, Thailand, Vietnam and Malaysia have lower $NH_3$ concentrations (typically <5 $\mu g\ m^{-3}$) for both model and measurements. Relative differences between model and measurement are generally small for the majority of sampling sites, and where they are large it is a consequence of expressing a difference relative to a small measured concentration. For example, the largest relative difference of 420%, which is in Vietnam, applies to a very small measured $NH_3$ concentration of 0.83 $\mu g\ m^{-3}$.

The modelled annual $NH_3$ concentrations at the NNMDN locations in China are slightly higher than the measurements (NMB = 0.29, Table 2), with 62% of the sites having positive model minus measurement differences. The sampling site with the largest positive difference is Zhumadian, where modelled $NH_3$ exceeds the measurement by 16.9 $\mu g\ m^{-3}$ (98% relative to measurement). The largest negative difference (−13.0 $\mu g\ m^{-3}$, −82% relative to measurement) is for the Wuwei site. The large concentration differences reflect the much larger $NH_3$ concentrations in China. By contrast, there is no significant difference

between model and measurement in most Southeast Asia countries. The average difference (mean bias) of annual $NH_3$ concentrations across all locations in the EANET network is 0.29 $\mu g\ m^{-3}$, which is a factor of ten smaller than the mean bias of 2.90 $\mu g\ m^{-3}$ for the NNDMN network.

Figure 6 and Table 2 also present the statistical relationships between modelled and measured annual average concentrations in China for $NO_2$, $NH_4^+$, $HNO_3$ and $NO_3^-$. Both $NH_3$ and $NO_2$ display strong linear relationships, while the

secondary species $NH_4^+$ and $NO_3^-$ show poorer correlations. The poorest agreement is for $HNO_3$ (Table 2). However, modelled $HNO_3$ concentrations agree much better with measurements in EANET and other networks (shown later), which suggests differences in measurement data among networks. Artefact-free measurement of $HNO_3$ is a known challenge (Tang et al., 2018b; Cheng et al., 2012; Sickles et al., 1999). The biases between model and NNDMN measurement are quite small for most species except for $HNO_3$. The overall annual average $NH_3$ concentrations are 13.0 and 10.1 $\mu g\ m^{-3}$ for model and

measurement respectively. The annual modelled network average $NO_2$ concentration of 28.6 $\mu g\ m^{-3}$ is only 22% greater than the measured network average $NO_2$ of 23.5 $\mu g\ m^{-3}$. The modelled and measured network average annual mean $NH_4^+$ concentrations are equal at 8.1 $\mu g\ m^{-3}$. The proportions of modelled and measured data that are within a factor of 2 are 75% for $NH_3$, 83% for $NO_2$, 78% for $NH_4^+$, and 71% for $NO_3^-$; the Fac2 for $HNO_3$ is, however, only 21%.

For comparisons at EANET sites, $NO_2$ has the highest correlation ($R$ = 0.84) amongst the gaseous species, followed by

$HNO_3$ ($R$ = 0.81), despite relatively higher biases between model and measurement ($NMB_{HNO3}$ = 1.13, $NMB_{NO2}$ = 0.67). The linear correlations are similar for $NH_3$ and $SO_2$, and both also exhibit small biases. The network-averaged modelled and measured annual average $NH_3$ concentrations are 1.92 $\mu g\ m^{-3}$ and 1.63 $\mu g\ m^{-3}$ respectively (NMB = 0.18). The equivalent data for $SO_2$ are 3.31 and 2.96 $\mu g\ m^{-3}$ (NMB = 0.12). For the aerosol components, the model simulates higher $NH_4^+$ concentrations (by 59%), but slightly lower $NO_3^-$ and $SO_4^{2-}$ concentrations (by 19% and 11%, respectively). Linear correlations of aerosol

components between model and EANET measurements are high ($R$ = 0.73-0.74). In summary, the model shows good performance in simulating key inorganic pollutants at EANET locations. The comparison statistics also show a systematically better model-measurement agreement for EANET than for NNDMN for all species.





### 3.2.2 Europe

The annual-mean NH₃ concentration map for Europe (Fig. 7) shows the highest NH₃ concentrations (>8 µg m⁻³) are in the Netherlands, Germany and Italy. Concentrations in northern Europe, such as Scandinavia, are smaller (<2 µg m⁻³), which is consistent with less anthropogenic activities and colder temperatures in this region. The model simulations of large NH₃ concentrations in the Po Plain in northern Italy arise from the large NH₃ emissions associated with intensive farming of pigs, cattle and poultry (Carozzi et al., 2013; Skjøth et al., 2011). In the UK, NH₃ concentrations generally display a decreasing

trend from south to north for both model and measurement although Northern Ireland is a relatively high NH₃ region as well. Most sites with NH₃ concentrations around or below 1 µg m⁻³ are in northwest Scotland, where modelled NH₃ concentrations are equally low.

Across all monitoring sites in Europe, 79% show positive differences for model minus measurement of annual NH₃. The monitoring locations with the largest differences (3.11-3.98 µg m⁻³, Fig. 7) are located in Germany and Switzerland, while

most sites with differences close to zero are situated in Norway, Sweden, Finland and Scotland. The site with the largest relative difference, Rannoch in the highlands of Scotland, has an extremely low measured concentration of 0.07 µg m⁻³ relative to the modelled concentration, also low), of 1.13 µg m⁻³. A number of sites which have negative model minus measurement differences are in southern England and eastern Northern Ireland. The largest model underestimation of NH₃ (−3.18 µg m⁻³) is at the Brompton site in England which also has the highest observed NH₃ concentrations for the UK. However, it is important

to note that the UK NAMN is a high spatial density NH₃ monitoring network, with many sites deliberately located near local emission sources of NH₃ (Tang et al., 2018a), which the global model grid-average cannot capture.

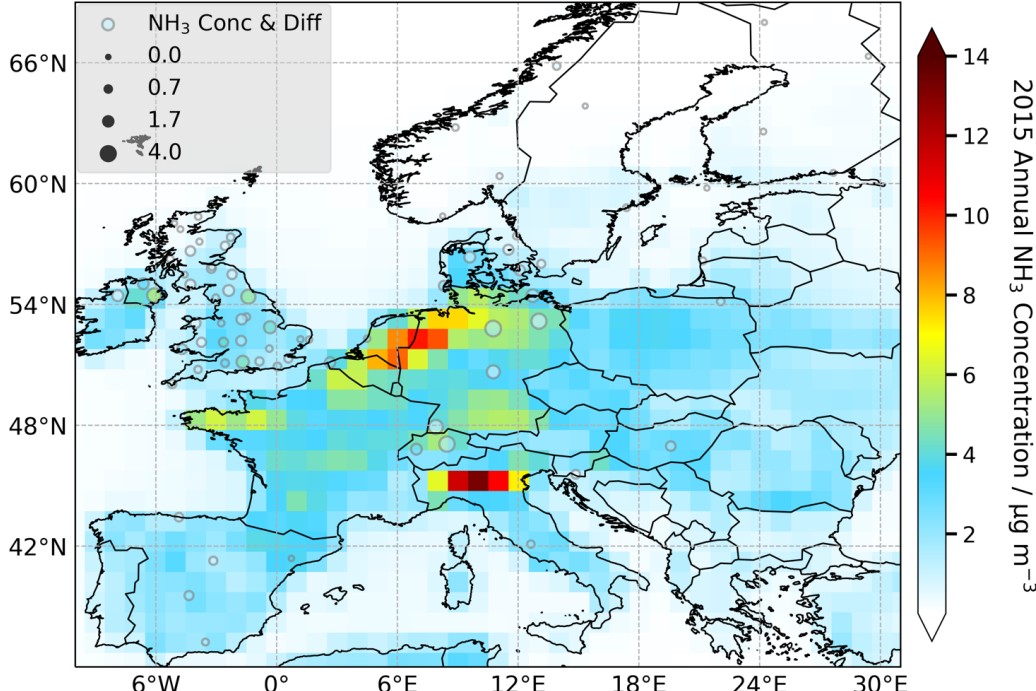

**Figure 7. Modelled and measured 2015 annual mean NH₃ concentrations in Europe. Measurements are from the UK NAMN and Europe EMEP/CCC networks. Monitoring sites are indicated by circles whose colour represents the measured concentration and whose diameter (see inset legend) represents the absolute difference between model and measurement.**





The linear relationships between model and measurement for 2015 annual average NH₃, NO₂, SO₂, NH₄⁺, NO₃⁻, and SO₄²⁻ concentrations in Europe are shown in Fig. 8 and a summary of the statistical comparisons is shown in Table 3. A few UK NAMN sites are part of the European EMEP/CCC network. Where a model grid contains multiple measurement sites, the average of the measured values is used.

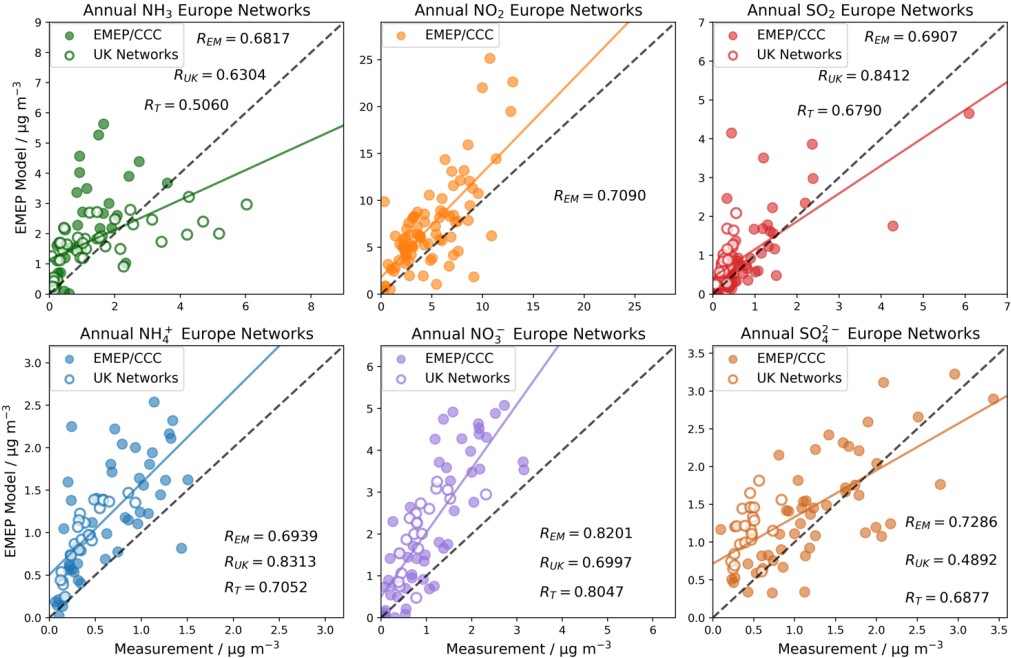

**Figure 8. Scatter plots of 2015 annual mean modelled and measured NH₃, NO₂, HNO₃, SO₂, NH₄⁺, NO₃⁻, and SO₄²⁻ concentrations at European monitoring network locations. In each plot, the solid line is the least-squares regression line and the black dashed line is the 1:1 line. $R_T$ is the overall correlation coefficient between model and all measurements.**

**Table 3. Summary statistics of model comparison with European measurements for 2015 (UK and EMEP/CCC networks). $N$ is the number of paired data of model and observation. $R$ is Pearson's coefficient. Fac2 is the fraction of data points within a factor of 2. Mean_O and Mean_M are annual average concentrations (µg m⁻³) of observation and model respectively. NMB is normalized mean bias, NME is normalized mean error.**

| Species | $N$ | $R_T$ | Fac2 fraction | Mean_O | Mean_M | NMB | NME |
|---------|-----|-------|---------------|--------|--------|------|------|
| NH₃ | 77 | 0.51 | 0.48 | 1.26 | 1.76 | 0.40 | 0.79 |
| NO₂ | 82 | 0.71 | 0.62 | 4.90 | 7.27 | 0.48 | 0.65 |
| HNO₃ | 48 | 0.60 | 0.65 | 0.38 | 0.31 | -0.18 | 0.50 |
| SO₂ | 90 | 0.68 | 0.57 | 0.65 | 0.90 | 0.39 | 0.70 |
| NH₄⁺ | 72 | 0.71 | 0.39 | 0.56 | 1.11 | 0.98 | 1.01 |
| NO₃⁻ | 69 | 0.80 | 0.42 | 1.09 | 2.18 | 0.99 | 1.05 |
| SO₄²⁻ | 75 | 0.69 | 0.65 | 1.02 | 1.34 | 0.32 | 0.51 |



There is a clear linear correlation between model and measurement for both primary and secondary species (Fig. 8). Correlation is highest for $NO_3^-$ ($R_T = 0.80$), followed by $NO_2$ and $NH_4^+$ ($R_T = 0.71$) and weakest for $NH_3$ ($R_T = 0.51$). However, the $NH_3$ data appear to be distributed into two groups, one characterized by positive model bias mainly associated with EMEP/CCC network locations, and one characterised by negative model bias mainly associated with the UK network. The former may be a result of overestimation of $NH_3$ in the emission inventory, the latter may be caused by UK measurement locations adjacent to agricultural $NH_3$ sources (Tang et al., 2018a). The model-measurement comparisons of other gaseous species ($NO_2$, $SO_2$ and $HNO_3$) all show better correlations ($R = 0.60$-$0.71$) and smaller differences (NME 0.50-0.70) in comparison with $NH_3$.

The modelled concentrations of secondary components, $NH_4^+$, $NO_3^-$, and $SO_4^{2-}$, all match well with the spatial variations of measurements, with $R_T$ varying from 0.69 to 0.80 (Fig. 8). All three components show higher modelled than measured concentrations, to varying degree. The network-averaged $NH_4^+$ concentrations are 1.11 and 0.56 µg m$^{-3}$ for model and measurement respectively. For $NO_3^-$, the modelled average concentration is 2.18 µg m$^{-3}$ which is around twice the measurement mean. In comparison with $NH_4^+$ and $NO_3^-$, $SO_4^{2-}$ shows a smaller NMB (0.32), and a larger Fac2 fraction (64%).

In conclusion, across Europe the model exhibits a good performance in simulating annual average concentrations and spatial variations of major inorganic air pollutants, but with an overestimation of secondary $NH_4^+$, $NO_3^-$, and $SO_4^{2-}$. The overall agreement between model outputs and ambient measurements in Europe networks is as good as that in EANET network.

### 3.2.3 United States and Canada

Modelled and measured 2015 annual average $NH_3$ concentrations and differences in North America are shown in Fig. 9. The Canadian NAPS network includes limited sampling sites for $NH_3$ and all of them are situated close to the border with the USA. Areas with the highest $NH_3$ concentration are located in the Midwestern United States according to the model, but there are only a few measurement locations in these regions. Annual average $NH_3$ measurements in North America vary from 0.39 µg m$^{-3}$ to 3.74 µg m$^{-3}$, while the model concentrations at those locations range from 0.13 µg m$^{-3}$ to 4.62 µg m$^{-3}$. The model generally simulates slightly higher $NH_3$ concentrations than measurements: 67% of the model-measurement differences are positive and the mean model bias is 0.48 µg m$^{-3}$. The modelled and measured concentrations of $NH_3$ in North America are comparable to those in Europe and much smaller than those in East Asia.





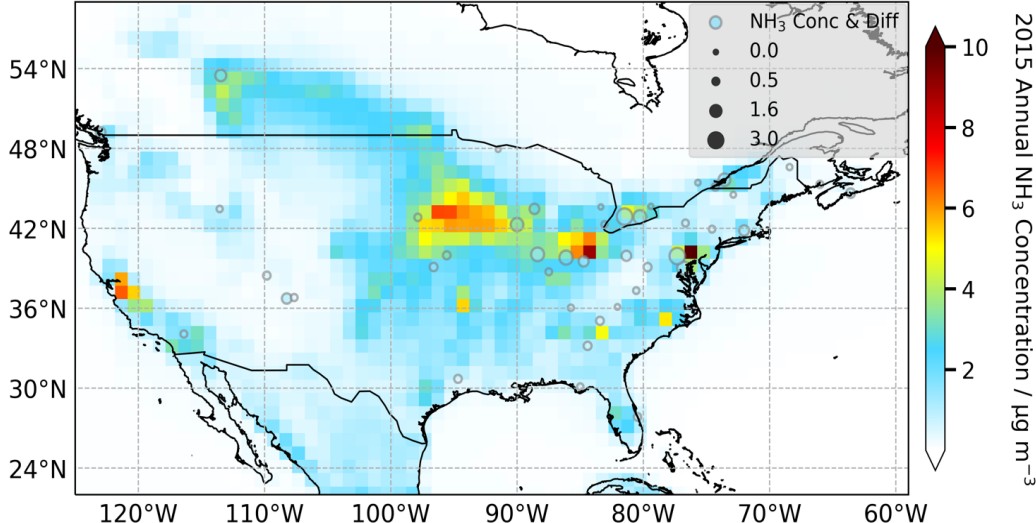


**Figure 9. Modelled and measured 2015 annual mean NH₃ concentrations in North America. Measurements are from the US AMoN Network and Canada NAPS Program. Monitoring sites are indicated by circles whose colour represents the measured concentration and whose diameter (see inset legend) represents the absolute difference between model and measurement.**

Figure 10 shows the linear relationships between model and measurement for 2015 annual average NH₃, NO₂, SO₂, NH₄⁺, NO₃⁻, SO₄²⁻ in North America. Table 4 provides the summary of statistical comparison metrics. The number of monitoring locations is greater than for the networks in East Asia, Southeast Asia, and Europe. The correlations between modelled and measured annual average NH₃, NO₂, HNO₃ concentrations in North America ($R_T = 0.59 - 0.72$) are similar to those in Europe and Southeast Asia, but the correlation for SO₂ is poor ($R_T = 0.27$). The reason for the poorer correlation between modelled

and measured SO₂ is unknown but may have a few causes: the emission inventory for SO₂ in North America may be too low, or some sampling sites may be set close to SO₂ point sources whilst grid-averaged model values are much lower. For the other three gaseous species the biases between model and measurement are in reasonable ranges. The network-averaged modelled NH₃ concentrations is 1.76 μg m⁻³ which is close to the measured average concentration of 1.28 μg m⁻³. For HNO₃, 78% of model data are within a factor of 2 of the measurements and the overall average concentrations are 0.51 μg m⁻³ and 0.39 μg

m⁻³ respectively (Table 4). Compared to NH₃ and HNO₃, the modelled annual NO₂ concentrations are generally smaller than measurements, leading to a negative NMB of −0.39.

    Clear linear relationships are observed between modelled and measured annual average concentrations for all three secondary pollutants (Fig. 10, Table 4), among which SO₄²⁻ has the highest correlation coefficient (0.86), the largest Fac2 (87%) and the smallest NMB and NME. This reflects excellent capability by the model to capture the spatial variation of SIA

constituents. In terms of absolute concentrations, modelled concentrations are on average higher than measured to varying degrees for NH₄⁺, NO₃⁻, and SO₄²⁻, as is the case in Europe. This may be due to gas-to-particle conversion process being too fast in the model or sinks of these secondary species being too small. The network-averaged NH₄⁺ concentrations are 1.06 μg m⁻³ and 0.50 μg m⁻³ for model and measurement respectively. For NO₃⁻ the equivalent concentrations are 1.19 μg m⁻³ and 0.58 μg m⁻³. Both NH₄⁺ and NO₃⁻ show relatively small Fac2 fractions due to model overestimation. By contrast, the smallest

differences are for SO₄²⁻ concentrations. The average model SO₄²⁻ concentration is 1.27 μg m⁻³ which only exceeds the average measurement concentration by 31%.



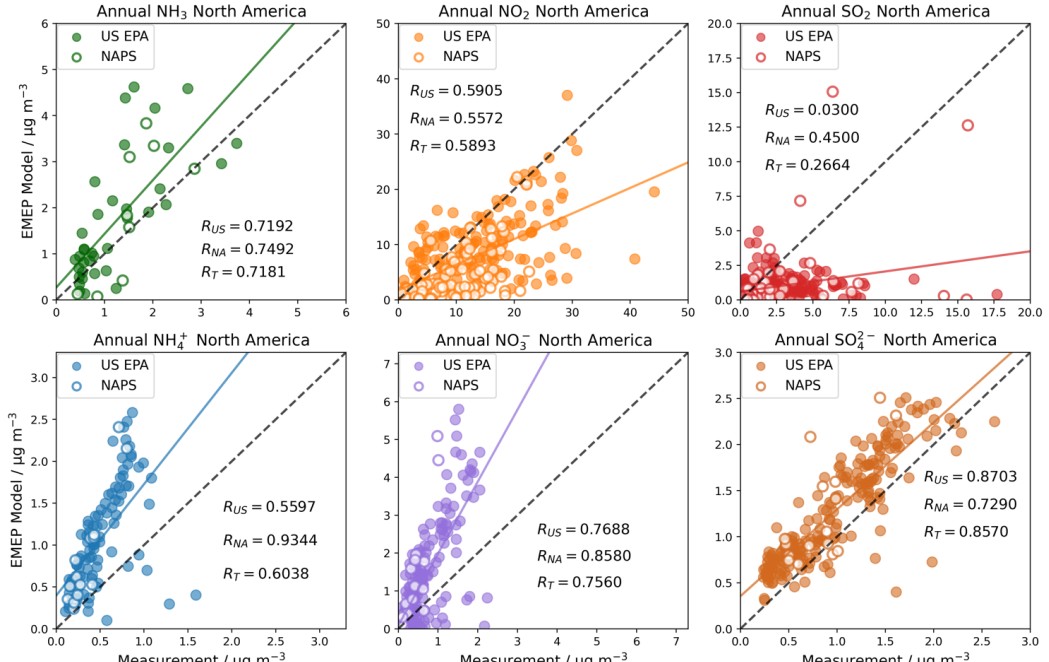

**Figure 10. Scatter plots of 2015 annual mean modelled and measured NH₃, NO₂, HNO₃, SO₂, NH₄⁺, NO₃⁻, and SO₄²⁻ concentrations at North American monitoring network locations. In each plot, the solid line is the least-squares regression line and the black dashed line is the 1:1 line. $R_T$ is the overall correlation coefficient between model and all measurements.**

**Table 4. Summary statistics of model comparison with measurements for 2015 in North America (USEPA and NAPS networks). $N$ is the number of paired data of model and observation. $R$ is Pearson's coefficient. Fac2 is the fraction of data points within a factor of 2. Mean_O and Mean_M are annual average concentrations (μg m⁻³) of observation and model respectively. NMB is normalized mean bias, NME is normalized mean error.**

| Species | $N$ | $R$ | Fac2 fraction | Mean_O | Mean_M | NMB | NME |
|---------|-----|-----|---------------|--------|--------|------|------|
| NH₃ | 45 | 0.72 | 0.64 | 1.28 | 1.76 | 0.37 | 0.57 |
| NO₂ | 259 | 0.59 | 0.55 | 12.27 | 7.43 | -0.39 | 0.49 |
| HNO₃ | 9 | 0.62 | 0.78 | 0.39 | 0.53 | 0.36 | 0.52 |
| SO₂ | 264 | 0.27 | 0.31 | 2.43 | 0.96 | -0.61 | 0.75 |
| NH₄⁺ | 106 | 0.60 | 0.32 | 0.50 | 1.06 | 1.12 | 1.25 |
| NO₃⁻ | 212 | 0.76 | 0.36 | 0.58 | 1.19 | 1.05 | 1.32 |
| SO₄²⁻ | 216 | 0.86 | 0.87 | 0.97 | 1.27 | 0.31 | 0.36 |

### 3.2.4 Comparison of temporal variation of modelled concentrations with measurements

The NNDMN, EANET, NAMN and EMEP/CCC monitoring networks also provide higher-temporal-resolution data, which allows a comparative assessment of monthly variations in model simulations (Fig. 11). As well as model-imposed temporal variations in emissions, the NH₃ concentrations are also driven by meteorological variations, in particular warmer temperatures favour partitioning of reduced N to gaseous NH₃. Missing measurement data for certain months and sites means the number of comparisons varies from one month to another.




**Figure 11. Monthly variations in modelled and measured NH₃ concentrations for locations in four monitoring networks in 2015. The box extends from the lower to upper quartile values of the data, with an orange line at the median and a green point at the mean. The whiskers represent 5% and 95% persentiles.**




In general, measurements of monthly average $NH_3$ concentrations in the China NNDMN show a trend of high in summer (mean: 14.6 µg m$^{-3}$, Table 5) and low in winter (mean: 6.54 µg m$^{-3}$). The seasonal pattern in the model simulations is slightly different, with dual peaks of $NH_3$ concentrations in March and August, but seasonal averages for spring and summer in model are similar to summer measurements at 14.6 µg m$^{-3}$ and 14.8 µg m$^{-3}$ respectively. Similar to measurements, the modelled $NH_3$

concentration is also lowest in winter (9.09 µg m$^{-3}$). For the EANET, both modelled and measured $NH_3$ median concentrations show a less clear varying trend than other networks, which might be due to the distributions of monitoring sites. A large number of sites in Southeast Asia are located in the tropics where the climate is characterised by a small temperature range and substantial rainfall, which leads to a very small range of fluctuations of $NH_3$ concentrations. The monthly averages indicate that measurements peak in April and October and are minimum in March and August, while the model has higher

concentrations in March, April, August and October, and lower concentrations in January and February. However, the fluctuation in the all-site monthly averages is small, ranging from 1.21 µg m$^{-3}$ to 3.21 µg m$^{-3}$, and from 1.77 µg m$^{-3}$ to 2.30 µg m$^{-3}$, for model and measurement respectively. The variation in monthly medians is even smaller.

**Table 5. Seasonal averages of monthly $NH_3$ concentrations (µg m$^{-3}$) for model (Mod) and measurements (Obs) in four monitoring networks. Spring: Mar, Apr, May; Summer: Jun, Jul, Aug; Autumn: Sep, Oct, Nov; Winter: Dec, Jan Feb.**

| Networks | Spring | | Summer | | Autumn | | Winter | |
|---|---|---|---|---|---|---|---|---|
| µg m$^{-3}$ | Obs | Mod | Obs | Mod | Obs | Mod | Obs | Mod |
| China | 10.9 | 14.6 | 14.6 | 14.8 | 8.72 | 12.9 | 6.54 | 9.09 |
| East Asia | 1.99 | 2.91 | 1.95 | 2.64 | 2.17 | 2.40 | 2.02 | 1.27 |
| UK | 1.94 | 1.71 | 1.43 | 2.49 | 1.36 | 1.90 | 1.10 | 0.61 |
| EMEP/CCC | 0.83 | 1.90 | 0.82 | 2.53 | 0.54 | 1.77 | 0.44 | 0.84 |

For the UK NAMN, both mean and median concentrations (Fig. 11) show that model and measurement exhibit higher $NH_3$ concentrations in spring and summer, and lower concentrations in winter. One small difference is in the timing of the $NH_3$ concentration maximum. The highest measured $NH_3$ concentrations are in spring, whereas modelled concentrations have a

maximum in summer. The differences between all-site monthly mean and median concentrations, and between the maximum and minimum values, in measurement are much larger than in the model, indicating a broad sub-grid variability that cannot be captured by the global model as the spatial averaging process smooths out these highly localised concentration gradients. For the European EMEP/CCC network, the model is in excellent agreement with measurement in respect of temporal pattern despite its higher absolute concentrations. Both model and measurement show a continuous period of higher $NH_3$

concentrations from spring to summer and lower $NH_3$ concentrations in autumn and winter.

Similar model-measurement monthly comparisons for $NH_4^+$ in 2015 are presented in Supplementary Materials (Fig. S6). Consistent monthly patterns are observed for both model and measurement in EANET, AGANet (UK) and EMEP/CCC networks: larger $NH_4^+$ concentrations are found in February, March and October, while the lowest concentration appears in July. For NNDMN locations, the model and measurement show a similar late summer peak but display inverse trend in winter

and spring.

In summary, the simulated concentrations of $NH_3$ and $NH_4^+$ and their month-to-month variability are generally in line with measurement data in most global regions despite the model resolution of 1° × 1°. The model comparisons with European measurements exhibit greater agreements than with East Asia and Southeast Asia measurements. The divergence in NNDMN


and EANET likely comes from shortcomings in the temporal profiles of emission inventories and is affected by the distribution
of limited measurement sites. A comparison of model outputs from STOCHEM-CRI and WRF-Chem-CRI with satellite
observations (Khan et al., 2020) also highlights a poor temporal agreement for NH₃ seasonality. Further model experiments
are required to investigate the impacts of different monthly emission and local meteorology on temporal variations of reduced
N species.

### 3.3 Comparisons of modelled precipitation and wet deposition with measurements

The evaluations of model performance for precipitation and wet deposition are based on the 4 monitoring networks (China,
East Asia, Europe and United States) that report both precipitation and precipitation concentration measurements for 2015.
The total annual wet deposition (WDEP) is calculated as,

$$WDEP = \bar{C} \times \sum P_i$$

where $\bar{C}$ (also referred to here as Prec Conc) is the precipitation-weighted annual average concentration

$$\bar{C} = \frac{\sum (C_i \times P_i)}{\sum P_i}$$

and $C_i$ is the concentration, and $P_i$ is the depth, of each individual precipitation event $i$ in the year. Prec Amount, $\sum P_i$, is the
total precipitation depth for the year. When $C_i$ (and $\bar{C}$) are expressed in units of mg L⁻¹, and $P_i$ in mm, then WDEP has units
of mg m⁻².

Figure 12 shows for each location in each of the four networks the comparisons between modelled and measured annual
precipitation, precipitation-weighted annual average concentration of reduced N (in the form of NH₄⁺) and annual total wet
deposition of reduced N in 2015. Table 6 summarises the statistical metrics associated with each comparison. The comparisons
of modelled and measured total rainfall show that the model is capable of simulating spatial variations of precipitation over
different global regions. The correlation coefficient $R$ ranges from 0.65 to 0.78 with an average of 0.72. The high Fac2
proportions indicate that the model can simulate the precipitation amount in EANET (82%), EMEP/CCC (91%) and US NTN
(82%) locations, but not so well for NNDMN (43%). In terms of model-measurement biases, the model underestimates annual
precipitation amounts by 13%-45%. Given the 1° spatial resolution of the model and the localised nature of precipitation
events, such a model underestimation range is expected.

The model performance in precipitation concentrations of reduced N varies between NNDMN and other networks. Whilst
comparisons for EANET, EMEP/CCC and US NTN show close to 1:1 linear relationships with $R$ values all >0.71, comparison
at NNDMN locations shows a relatively poor correlation ($R$ = 0.45). This may reflect instrumental and experimental
differences between monitoring networks. Considering the limited number of monitoring sites in NNDMN, more measurement
data are required to draw a more representative model-measurement comparison in China.

The measured annual wet deposition of reduced N is affected by the quality of the measurement of both collected rainfall
and precipitation-weighted average NH₄⁺ concentration. Based on measurement locations, NNDMN shows the largest annual
reduced N wet deposition for both model (777 mgN m⁻²) and measurement (986 mgN m⁻²), followed by EANET (model 380
mgN m⁻², measurement 499 mgN m⁻²), EMEP/CCC (model 146 mgN m⁻², measurement 226 mgN m⁻²) and US NTN (model
135 mgN m⁻², measurement 192 mgN m⁻²). The model simulates lower total reduced N wet depositions by 21% - 35% across
the four networks. This general model underestimation is largely driven by the underestimation of total precipitation, and to





585 less extent the precipitation concentration. Across the four networks, agreement between modelled and measured wet deposition of reduced N is best for the US NTN with $R = 0.75$ and Fac2 = 81%.

The comparison for global wet deposition of total oxidized N (in the form of $NO_3^-$) exhibits similar results and is presented in the Supplementary Material (Fig. S7 and Table S1). The modelled precipitation-weighted concentrations of $NO_3^-$ has relatively good agreements with measurements in EANET, EMEP/CCC and US NTN networks with $R$ ranging from 0.69 to 590 0.80, while the comparisons in NNDMN show a poorer linear correlation between model and measurement ($R = 0.40$). In terms of biases, the model tends to simulate higher $NO_3^-$ concentrations in precipitations in EANET (NMB = 0.52) and US NTN (NMB = 1.04) networks but underestimate in NNDMN (NMB = -0.37). In general, the greatest model-measurement agreement for oxidized N wet deposition is found in US NTN, followed by EMEP/CCC and EANET, and to the least extent NNDMN, which again suggests systematic differences between monitoring networks rather than issues with the modelling of 595 atmospheric chemistry and meteorology.

On the whole, the modelled reduced and oxidized N show similar linear relationships with measurements in precipitation and wet deposition in all regions, which further supports the utilization of the WRF and EMEP MSC-W modelling system to investigate $N_r$ processes globally.

600 **Table 6. Summary statistics of model comparison with measurements for annual precipitation (Prec Amount, mm), precipitation-weighted mean concentration of $NH_4^+$ (Prec Conc, mgN L$^{-1}$), and wet deposition of reduced N (WDEP, mgN m$^{-2}$) in 2015. $N$ is the number of measuring sites. $R$ is Pearson's coefficient. Fac2 is the fraction of data points that are within a factor of 2. Mean_O and Mean_M of Prec Conc are annual averages of observation and model respectively. Mean_O and Mean_M of Prec Amount and WDEP are annual totals. NMB is normalized mean bias, NME is normalized mean error.**

| Networks | Variables | $N$ | $R$ | Fac2 fraction | Mean_O | Mean_M | NMB | NME |
|---|---|---|---|---|---|---|---|---|
| China | Prec Amount | 21 | 0.73 | 0.43 | 913 | 502 | -0.45 | 0.49 |
| | Prec Conc | 21 | 0.45 | 0.71 | 2.00 | 2.18 | 0.09 | 0.45 |
| | WDEP | 21 | 0.59 | 0.62 | 986 | 777 | -0.21 | 0.42 |
| East Asia | Prec Amount | 50 | 0.65 | 0.82 | 1585 | 1270 | -0.20 | 0.39 |
| | Prec Conc | 44 | 0.71 | 0.66 | 0.44 | 0.42 | -0.04 | 0.62 |
| | WDEP | 40 | 0.68 | 0.62 | 499 | 380 | -0.24 | 0.50 |
| Europe | Prec Amount | 101 | 0.78 | 0.91 | 863 | 749 | -0.13 | 0.31 |
| | Prec Conc | 89 | 0.77 | 0.85 | 0.32 | 0.26 | -0.19 | 0.28 |
| | WDEP | 89 | 0.61 | 0.75 | 226 | 146 | -0.35 | 0.41 |
| United States | Prec Amount | 206 | 0.73 | 0.82 | 1030 | 690 | -0.33 | 0.39 |
| | Prec Conc | 207 | 0.76 | 0.90 | 0.22 | 0.22 | 0.03 | 0.30 |
| | WDEP | 206 | 0.75 | 0.81 | 192 | 135 | -0.30 | 0.36 |

605



**Figure 12. Scatter plots of model-measurement comparisons of 2015 annual wet deposition variables for reduced N (in the form of NH₄⁺) for four measurement networks: NNDMN, EANET, EMEP/CCC and US NTN. Left panels are annual precipitation. Middle panels are precipitation-weighted annual average NH₄⁺ concentration in precipitation. Right panels are annual total wet deposition of NH₄⁺. In each plot, the coloured line is the least squares regression, and the black dashed line is the 1:1 line**




**4 Discussion**

The work presented here is motivated by the use of the EMEP MSC-W-WRF model for global-scale analyses of atmospheric nitrogen and SIA chemistry, fluxes and budget, particularly species that contain reduced N (i.e. gaseous $NH_3$ and particulate $NH_4^+$). The model evaluation, conducted both spatially and temporally, is based on the available data in 2010 and 2015 from 9 monitoring networks that span the range of ambient measurements in East Asia, Southeast Asia, Europe, and North America.

Table 7 summarises the global comparison between model and surface measurements in 2015. The correlation coefficients ($R$) between modelled and measured concentrations of most species (i.e. $NH_3$, $NO_2$, $NH_4^+$, $NO_3^-$ and $SO_4^{2-}$) are all greater than 0.78 except for $HNO_3$ and $SO_2$. The wet deposition of reduced N shows a stronger linear correlation ($R = 0.78$) than oxidized N ($R = 0.64$). For reduced N species, the evaluation shows that the model overestimates $NH_3$ and $NH_4^+$ worldwide with a NMB of 31% and 37% respectively. For oxidized N species, the NMB values for $NO_2$ and $NO_3^-$ are 23% and 61% and, in contrast, $HNO_3$ is underestimated by 34%. Slightly higher concentrations are also simulated by the model worldwide for both $SO_2$ and $SO_4^{2-}$ with a NMB of 10% and 21% respectively. For wet deposition, the model has smaller values on average for reduced N (NMB = −29%) but larger values for oxidized N (NMB = 26%). Given the intrinsic discrepancies between local site measurement and a global-scale chemistry model grid, these comparisons are good and are comparable with model evaluation statistics determined for models of similar resolution (Hauglustaine et al., 2014; Bellouin et al., 2011; Pringle et al., 2010; Xu and Penner, 2012).

**Table 7. Summary statistics of global model evaluation of atmospheric concentrations (µg m$^{-3}$), annual precipitation (Prec Amount, mm), precipitation-weighted mean concentration of $NH_4^+$ and $NO_3^-$ (Prec Conc, mgN L$^{-1}$), and wet deposition (mgN m$^{-2}$) of reduced N (RDN) and oxidized N (OXN), in 2015. $N$ is the number of measuring sites. $R$ is Pearson's correlation coefficient. Fac2 is the fraction of data points that are within a factor of 2. Mean_O and Mean_M of Prec Conc are annual averages of observation and model respectively. Mean_O and Mean_M of Prec Amount and WDEP are annual totals. NMB is normalized mean bias, NME is normalized mean error.**

| Globe | Variables | $N$ | $R$ | Fac2 fraction | Mean_O | Mean_M | NMB | NME |
|---|---|---|---|---|---|---|---|---|
| Atmospheric concentration | $NH_3$ | 173 | 0.85 | 0.57 | 2.55 | 3.35 | 0.31 | 0.63 |
| | $NO_2$ | 372 | 0.78 | 0.62 | 7.43 | 9.11 | 0.23 | 0.56 |
| | $HNO_3$ | 109 | 0.54 | 0.50 | 1.44 | 0.95 | -0.34 | 0.68 |
| | $SO_2$ | 390 | 0.61 | 0.45 | 1.05 | 1.16 | 0.10 | 0.82 |
| | $NH_4^+$ | 230 | 0.83 | 0.43 | 1.31 | 1.80 | 0.37 | 0.69 |
| | $NO_3^-$ | 334 | 0.83 | 0.40 | 1.41 | 2.28 | 0.61 | 0.88 |
| | $SO_4^{2-}$ | 320 | 0.83 | 0.81 | 1.17 | 1.42 | 0.21 | 0.38 |
| Wet deposition (RDN) | Prec amount | 377 | 0.73 | 0.82 | 1052 | 772 | -0.27 | 0.38 |
| | Prec conc | 360 | 0.80 | 0.85 | 0.37 | 0.36 | -0.01 | 0.39 |
| | WDEP | 355 | 0.78 | 0.76 | 278 | 198 | -0.29 | 0.41 |
| Wet deposition (OXN) | Prec conc | 366 | 0.79 | 0.58 | 0.31 | 0.35 | 0.16 | 0.58 |
| | WDEP | 363 | 0.64 | 0.60 | 171 | 215 | 0.26 | 0.72 |

Both model and measurement have uncertainty that constrains the extent to which statistical analyses between modelled and measured data can be utilized to assess a model's performance. A reliable evaluation of a model requires a high quality of measurement as well. For instance, sampling and chemical analysis procedures such as the instrument calibration, the choice of sampling filters/tubes, the storage, extraction and chemical speciation of air samples all have different uncertainties propagated to the final measured variable. In particular, this study and the above-mentioned global modelling studies all show difficulties in representing surface $NO_3^-$ and $NH_4^+$ concentrations, which are currently overestimated by around a factor of 2 in Europe and North America. Such positive biases between modelled and measured $NO_3^-$ and $NH_4^+$ are speculated to be partially associated with negative sampling artifacts in measurements as evaporation of $NH_4NO_3$ from sampling filters has been reported to cause losses of up to 50% in summer conditions (Hauglustaine et al., 2014; Vecchi et al., 2009; Yu et al., 2005). Further work is required to better characterize and quantify the uncertainty of individual $NO_3^-$ and $NH_4^+$ measurements. In general, the relative measurement uncertainty increases markedly as concentration decreases (Thunis et al., 2013; Pernigotti et al., 2013). The EMEP/CCC data report for 2015 estimates a combined sampling and chemical analysis uncertainty range of 15-25% (Hjellbrekke, 2017), while the detailed uncertainty information in other monitoring networks is not publicly available.

Similarly, different input, configurations and computing processors also have influences on the model output, and the quantification of such influence is rather complicated (Kong et al., 2020). The choice of emission input is a good example (Aleksankina et al., 2019). The compilation of an emission inventory is partially based on reported measurement data and partially on expert estimation, which consequently leads to a certain uncertainty in emission magnitudes and temporal profiles (EMEP/EEA, 2019; Hilde Fagerli, 2017; Klimont et al., 2017; Wiedinmyer et al., 2011; Zheng et al., 2012). The completeness and consistency of submitted emission data differs significantly across countries as well. As discussed in Section 3.1, the two global emission inventories used in this work, HTAP and ECLIPSE$_E$, have shown large localised discrepancies in $NH_3$, $NO_2$ and $SO_2$ emissions in certain world regions, which is presumably ascribable to the inclusion or exclusion of a particular local point source in the compilation process. The influence of these discrepancies on model-simulated surface concentration differs in terms of primary or secondary component and varies from one region to another, although such greatly localised influences are diminished during the spatial averaging processes. It is therefore important to acknowledge that the performance of any model is subject to the quality of model input data which includes not only emissions but also meteorology and other aspects of model parameters. Moreover, no one can guarantee error-free models; in the same way that observations are likely to be not error free. Often in the atmospheric modelling community these potential model errors are not discussed or acknowledged.

Aside from intrinsic uncertainties in model and/or measurement values, the model and measurement may also not agree concerning the averaging time periods and the diameters of the sampled particles. A certain number of measurements may be missing from a time series due to unpredictable instrument failure and/or because the measurement averaging period does not match the model averaging time period. It is clear that the sampling time and size distributions of measurements vary from one monitoring network to another, and from species to species. For example, in Canada, $NH_4^+$ concentrations within $PM_{2.5}$ are measured, while the particle size cut-off for the DELTA system used in the UK and China is around 4.5 μm (Tang et al., 2018a; Tang et al., 2018b; Xu et al., 2019). The modelled $NH_4^+$, $SO_4^{2-}$, and fine $NO_3^-$ are all in $PM_{2.5}$. Another example is that in the US and Canada gaseous species like $NO_2$ and $SO_2$ are monitored continually throughout the year and thus the corresponding annual average concentrations are calculated in the same way as the model, whilst the aerosol components such as $NO_3^-$ and $NH_4^+$ are measured once per 6 days (or once per week). From this point of view, there is an inherent divergence already in comparing the model simulations with ambient measurements.

Even if both model and measurement were perfect representations, there still would not be complete agreement because a measurement is for a single point in space whereas, even for models with high spatial resolution, model output is a volume average. For a global model simulation with grid resolution of 1° × 1°, the monitoring site simply samples the air in one part


of that grid volume and at a specific height above the ground, which may often not reflect the average concentration for the grid. Indeed, there are particular monitoring sites where measurements are exceedingly affected by local sources. The UK NAMN is a good example, in which quite a few sites are purposely set near agricultural sources and therefore yield higher $NH_3$ concentrations than model grid-average predictions. The US EPA also has many monitors set up next to roads with heavy traffic and hence observed much higher $SO_2$ levels. The representativeness of an urban (or rural) site for the air in the

corresponding model grid will therefore depend on the relative size of that specific urban (or rural) area within that model grid.

The intention here is to provide an overview of how the EMEP-WRF model-measurement agreements vary among different monitoring networks and among different chemical species for evaluation of a chemistry transport model in a global context. In general, the model shows better linear correlations with surface concentration measurements in East Asia ($\bar{R} = 0.73$ over 7 species), Europe ($\bar{R} = 0.67$ over 7 species) and North America ($\bar{R} = 0.63$ over 7 species) than in China ($\bar{R} = 0.35$ over 5

species). More specifically, comparisons in China show the model performs better in computing concentrations of primary pollutants (i.e. $NH_3$ and $NO_2$) than secondary species (i.e. $NH_4^+$ and $NO_3^-$), while the model evaluation statistics in East Asia, Europe and North America show almost equally good results over all species. This implies potential discrepancies in the measurements or emissions in China rather than general issues with meteorological and atmospheric chemistry modelling. The values of statistical metrics in this work are as good as other global model evaluation studies. A global model aerosol simulation

study (Hauglustaine et al., 2014) reported that the $R$ of global model results (LMDz-INCA global chemistry–aerosol–climate model, 1.9° latitude × 3.75° longitude resolution) versus measurements in 2006 for surface concentrations of $SO_4^{2-}$, $NH_4^+$ and $NO_3^-$ ranged 0.43-0.58 in Europe and 0.54-0.77 in North America, which is similar to our results presented here. The AeroCom phase III global nitrate experiment, which includes 9 models, reported slightly lower $R$ ranges than here for annual $NO_3^-$ in 2008: 0.081-0.735 in North America, 0.393-0.585 in Europe, and 0.226-0.429 in Southeast Asia (Bian et al., 2017); and the

agreements between model and observation for gas tracers in that study were even lower than here.

This work has utilized the EMEP MSC-W v4.34 coupled with WRF v3.9.1.1 model. As discussed above, model-measurement comparison statistics will vary in different global models to different extent. However, the broad discussions associated with fundamental differences between localised measurement and grid-volume averaged model output, unmatched temporal coverage, relatively higher uncertainties of emissions, and intrinsic limitations of measurement, are generalizable, as

ACTMs and other climate models are constructed similarly. Allowance for these inherent model-measurement discrepancies and uncertainties yield significantly less stringent requirements on acceptable model evaluation statistics than might initially be expected. Urban dispersion models (Denby et al., 2020; Hood et al., 2018) with higher resolutions have stronger capabilities of representing point sources and concentration gradients but are constrained even more by the accuracy of localised emission inventories and boundary conditions in the meantime, and therefore are only configured at an individual urban area. Global-

scale model simulation as presented here, in spite of acknowledged limitations on coarser spatial resolution, has the advantage of generating self-consistent chemistry fields and competence for investigating contemporary and potential future global reactive nitrogen and SIA atmospheric chemistry and their regional variations.

## 5 Conclusions

This model versus measurement study is motivated by the first application of a global version of the EMEP MSC-W model

with WRF meteorology (1° × 1° horizontal resolution) to study global reactive N and S chemistry and deposition. A comprehensive spatial and temporal comparison of model output against 9 monitoring networks from 4 world regions (East Asia, Southeast Asia, Europe and North America) has been undertaken, with a focus on the atmospheric concentrations and wet deposition of major inorganic pollutants, and on reduced nitrogen components in particular. Simulations were performed



with EMEP MSC-W model version 4.34 with WRF 3.9.1.1 meteorology, using both ECLIPSE$_E$ (2010 and 2015) and HTAP (2010 only) emission inventories. (ECLIPSE$_E$ refers to ECLIPSE annual emissions with EDGAR monthly profiles.)

In general, simulations of annual surface concentrations of a primary pollutant such as $NH_3$ are somewhat sensitive to the choice of HTAP or ECLIPSE$_E$ emission inventories in places where regional differences in primary emissions between the two emission inventories are apparent, e.g. China. By comparison, the impact of difference between the emissions inventories on concentrations of secondary species such as $NH_4^+$ is much smaller. The difference in 2010 global area-weighted annual average $NH_3$ concentration is 0.05 μg m$^{-3}$ (HTAP: 0.26 μg m$^{-3}$; ECLIPSE$_E$: 0.31 μg m$^{-3}$) which is 18% of the absolute concentration, whilst the $NH_4^+$ concentration difference is only 0.02 μg m$^{-3}$ or only 3.5% of the global average concentrations (HTAP: 0.316 μg m$^{-3}$; ECLIPSE$_E$: 0.305 μg m$^{-3}$). In terms of temporal profiles, the monthly average emissions vary similarly throughout the year in the four world regions after the monthly profiles derived from EDGAR are applied to the ECLIPSE annual total emissions.

Comparisons of 2010 and 2015 annual average concentrations between model and measurement demonstrate that the model captures well the overall spatial and temporal variations of major inorganic pollutants despite spanning large concentration ranges in different world regions. The discussion of model evaluation statistics mainly focuses on 2015 as the results for 2010 are similar.

In general, capturing correlation is more important than bias given the intrinsic discrepancies and uncertainties between the modelled and measured variables. In this work the model shows better linear correlations with measurement networks in Southeast Asia (Mean $R$ for 7 species: $\overline{R_7} = 0.73$), Europe ($\overline{R_7} = 0.67$) and North America ($\overline{R_7} = 0.63$) than in China ($\overline{R_5} = 0.35$ over 5 species), which implies potential discrepancies with some measurements and emissions rather than issues with modelling meteorological and atmospheric chemistry processes. Model-measurement bias varies from one species to another in different networks. $NH_4^+$ and $NO_3^-$ are the species overestimated the most by the model in Europe and North America but not so much in East Asia and Southeast Asia networks, reflecting that the model production of the two species might be too fast and/or the chemical and physical losses might be too slow in the two regions. The model performs the best in simulating $SO_4^{2-}$ concentrations in North America regarding overall statistics among various species in all networks.

Both model and measurement exhibit higher $NH_3$ concentrations in spring and summer, and lower concentrations in winter. The greatest agreement of temporal profile for model and measurement is found in Europe. The fluctuation of monthly average $NH_3$ concentrations in Southeast Asia throughout the year 2015 is fairly small for both model and measurement and the temporal trend is therefore less clear. Small differences appear regarding the specific peak concentration months in China and UK. Measurements in China show highest monthly concentration in July, while the model simulates two peaks in August and March. Highest $NH_3$ concentrations in the UK network are in spring, whereas the modelled concentrations peak in summer. Such disagreements again reflect the likelihood that the major driver of model discrepancies is the inaccuracy of temporal profiles of emissions rather than the simulation of atmospheric chemistries and physics.

The evaluation of wet deposition shows that the model is capable of simulating spatial variation of annual precipitation correctly in all four world regions (0.65-0.78 $R$ range) despite a 13-34% underestimation. Given that the spatial and temporal averaging smooths out highly localised effects of precipitation event, such model-measurement discrepancy is reasonable. In respect of the weighted precipitation concentrations, high linear correlations between measured and modelled $NH_4^+$ and $NO_3^-$ concentrations are observed in Southeast Asia, Europe and North America but not China, which may again suggest systematic difference among measurement rather than model. In general, the model shows the greatest consistency of annual total wet deposition with measurements in North America ($R$: 0.75 and 0.81 for reduced and oxidised N respectively; similarly, hereinafter), followed by Southeast Asia ($R$: 0.68 and 0.51), Europe ($R$: 0.61 and 0.64) and China ($R$: 0.59 and 0.13).



Despite discussed limitations in model-measurement comparisons, the detailed evaluations presented here support the utilization of this global implementation of the EMEP MSC-W-WRF coupled model for analyses of surface concentrations and wet depositions of major reactive N and S species in different world regions. Modelling of atmospheric chemistry and transport on the global scale has the advantage of providing consistent data with comprehensive spatial and temporal coverage, of filling in the research gap in global model evaluation, and of facilitating investigation of global reactive N and SIA deposition budgets and chemistry/policy-oriented model experiments for potential future scenarios.

**Code and data availability**

As described and referenced in Section 2 of this paper, this study used two open-source global models: the European Monitoring and Evaluation Programme Meteorological Synthesizing Centre -West atmospheric chemistry transport model (EMEP MSC-W, version 4.34, source code available at https://doi.org/10.5281/zenodo.3647990) and the Weather Research and Forecast meteorological model (WRF, version 3.9.1.1, www.wrf-model.org, http://dx.doi.org/10.5065/D68S4MVH). The two global emission inventories applied are described in Sect. 2.1. All measurement datasets used in this work are publicly available and their individual websites are listed in Sect. 2.2. The model and measurement output presented in figures and tables in this paper and the corresponding Python scripts are available at https://doi.org/10.5281/zenodo.5037080.

**Author contribution**

MH, DS and MV conceptualised and supervised the study. MV and PW contributed to model development and set-up and provided modelling support. MV provided computing resource. YG contributed to study design, undertook all model simulations, compilation of measurement datasets, formal data analyses, visualisation of the results and data curation, with discussion and refinement by all authors The original draft of the paper was written by YG with editing by MH. All authors provided review comments and approval of the final version.

**Competing interests**

The authors declare that they have no conflict of interest.

**Acknowledgments**

Y. Ge gratefully acknowledges studentship funding from the University of Edinburgh and its School of Chemistry. This work was in part supported by the UK Natural Environment Research Council (NERC), including grant nos. NE/R016429/1 and NE/R000131/1 as part of the UK-SCAPE and SUNRISE programmes delivering UK National Capability, and the European Modelling and Evaluation Programme under the United Nations Economic Commission for Europe Convention on Long-range Transboundary Air Pollution.



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
