# Peer review of "Evaluation of global EMEP MSC-W (rv4.34)-WRF (v3.9.1.1) model surface concentrations and wet deposition of reactive N and S with measurements"

_Geoscientific Model Development, 2021_

## Author Comment (AC1)

**gmd-2021-166: Evaluation of global EMEP MSC-W (rv4.34)-WRF (v3.9.1.1) model surface concentrations and wet deposition of reactive N and S with measurements**
**Ge et al.**

**Response to Reviewer #1**

We thank the reviewer for the time spent reading our manuscript. Below we include all the reviewer comments and provide in blue text our point-by-point responses.

Atmospheric chemistry and transport models (ACTMs) are crucial to understanding sources and impacts of reactive nitrogen ($N_r$) chemistry and its potential mitigation. In the manuscript, the authors undertook the first evaluation of the global version of the EMEP MSC-W ACTM driven by WRF meteorology ($1°×1°$ resolution), with a focus on surface concentrations and wet deposition of N and S species relevant to investigation of atmospheric Nr and secondary inorganic aerosol (SIA). The results of model-measurement comparisons, conducted both spatially and temporally, covering 9 monitoring networks worldwide, showed an overall nice agreement between simulated and observed data. The authors found that simulations of primary pollutants (e.g. $NH_3$) are sensitive to the choice of different inventories of primary emissions (e.g. China), but much less so for secondary components (e.g. $NH_4^+$). Furthermore, comparisons of 2010 and 2015 surface concentrations between model and measurement demonstrated that the model captured well the overall spatial and seasonal variations of the major inorganic pollutants, and their wet deposition in different regions worldwide. The model showed better correlations with annual average measurements for networks in Southeast Asia, Europe, and North America than in East Asia (data for 2015), suggesting potential issues with the measurements in the latter network. Temporally, both model and measurement agree on higher $NH_3$ concentrations in spring and summer, and lower concentrations in winter. The authors also reported high correlations between measured and modelled $NH_4^+$ precipitation concentrations in all regions except East Asia (receiving greater anthropogenic activities). They evaluated model-measurement bias for various atmospheric $N_r$ species in different networks as well. The greater uniformity in spatial correlations than in biases suggested that the major driver of model-measurement discrepancies were shortcomings in absolute emissions rather than in modelling the atmospheric processes. In summary, this study supported the application of this model framework for global analysis of current and potential future budgets and deposition of Nr and SIA. The authors' work provides strong implications of modelling the atmospheric processes regionally and globally, but the key point is the (relatively) accurate emission inventory of Nr species. The manuscript fits well the scope of this journal and merits to be published after minor revisions as follows.

Response: We thank the reviewer for their support of our work and for their recommendation of publication after attention to some minor revisions.

L80: wet deposition of Nr

Response: Requested change made.

L109-110: use the abbreviation of reference expression by "(Vieno et al., 2010; 2014; 2016)"

Response: Requested change made.

L127-128: As you mentioned here, all inventories were aggregated to $1° \times 1°$ resolution internally in the model. Could you make some comments on the uncertainties due to this aggregation (e.g. from resolutions of 0.1×0.1o and 0.5×0.5o)? I have the same concern on uncertainties for the re-assigned 11 selected nomenclatures for sources of air pollution sectors in all inventory emission sector-layers.

Response: The inventory spatial aggregation used the standard EMEP model data conservation aggregation. The re-assignment of emission sectors was achieved by directly summing the emissions from those emission inventory subsectors that are linked to a given SNAP sector to obtain the total emission from this SNAP sector. The uncertainty for this re-assignment process is therefore subject to the propagation rules for uncertainty calculation, which requires information on the uncertainties in all the individual subsector emissions. This information we do not have. We fully appreciate that compiling emission inventories, especially on a global scale, is tremendously complex and challenging, and that we definitely do not have better insight into uncertainties than do the emissions specialists. It is therefore not appropriate for us to attempt our own quantification of sectorial (and total) emissions uncertainties, which would amount only to guesswork.

However, what we can confirm is that our re-assignment of emission sectors does not have any material impact on model output of global total emissions. For instance, the 2010 annual global emission of $NH_3$ from the ECLIPSE emission inventory is 61.66 Tg $yr^{-1}$, while the calculated total emission from model output is 61.72 Tg $yr^{-1}$, a relative difference of <0.1%. Similarly, the 2010 global $NH_3$ emission from the HTAP emission inventory is 53.46 Tg $yr^{-1}$, and the same total from model output is 53.51 Tg $yr^{-1}$, which is also <0.1% discrepancy.

L966: Delete "14, 16(2014-08-21)" before "14, 8435-8447" and add the "doi" number before "2014".

Response: Requested change made.

---

## Author Comment (AC2)

**gmd-2021-166: Evaluation of global EMEP MSC-W (rv4.34)-WRF (v3.9.1.1) model surface concentrations and wet deposition of reactive N and S with measurements**

**Ge et al.**

**Response to Reviewer #2**

We thank the reviewer for the time spent reading our manuscript. Below we include all the reviewer comments and provide in blue text our point-by-point responses.

**Overarching comment:**

The manuscript provides a comprehensive evaluation of reactive nitrogen concentrations and wet deposition simulated by the WRF/EMEP-MSC-W modelling system against a suite of surface measurement networks over Europe, Asia, and North America. The goal of the manuscript is to establish the credibility of this modelling system in simulating the atmospheric transport and fate of reactive nitrogen on global scales for studying processing affecting concentrations and quantifying the impacts of policy scenarios. The results suggest that the performance of the modelling system is on par with the performance of other contemporary global-scale modelling systems. While I do not expect this general conclusion to change, there are several aspects of the manuscript that should be strengthened prior to its final publication as detailed in my main and specific comments below. The manuscript is well written and structured clearly, the presentation quality of all figures and tables is good, and references to other published studies on similar topics are provided where appropriate. I enjoyed reading Section 4 that provides a nice discussion of the inherent limitations of using surface measurements as criterion for establishing model credibility.

Response: We thank the reviewer for their supportive comments in general for our work.

**Main comments:**

1. It is not clear to me whether the authors used the available observations at their native temporal resolution (e.g. hourly, daily, weekly) and matched these native-resolution observations to the corresponding model values (for non-missing hours / days / weeks) to compute observed and modeled annual averages for the subsequent analyses, or whether they relied on pre-computed observed annual averages provided by some or all of the networks and then matched them to modeled annual averages (computed from all 8760 hours in a given year). Section 2.2 and the discussion in Section 4 suggest the latter approach was taken, but given that this approach can lead to temporal mismatching as discussed in Section 4, I don't see why such a choice would have been made. At least for the North American and

European networks I am familiar with (EMEP, NAPS, NTN, AMoN, and EPA Air Data), native resolution data is readily available (e.g. https://aqs.epa.gov/aqsweb/airdata/download_files.html, http://nadp.slh.wisc.edu/data/NTN/ntnAllsites.aspx, https://data-donnees.ec.gc.ca/data/air/monitor/national-air-pollution-surveillance-naps-program/Data-Donnees/2010/ContinuousData-DonneesContinu/HourlyData-DonneesHoraires/?lang=en) and should be used in the analysis to eliminate this factor of uncertainty.

Response: This study was concerned with a comparison between model and observations at annual and monthly time resolution on a global scale. Where the measurement databases supplied annual and/or monthly average values we used these data for our model-observation comparisons (or we averaged the monthly values where no annual average was supplied directly). We did this on the basis that the organisations undertaking the measurements are the experts on their measurements and are therefore the appropriate authority for determining valid temporally-averaged values to provide to the user community. Where annual (or monthly) average measurements were not provided in the database we undertook the averaging ourselves. We followed a standard pragmatic approach. An annual (or monthly) average was computed when ≥75% of the underlying data were available, otherwise the average was not computed. The 75% data capture criterion for atmospheric composition measurement is widely applied (in Europe at least), being laid down in pan-European air quality legislation and used by EMEP MSC-W model developers in their regular model status reports (Fagerli et al., 2019; Fagerli et al., 2011; Berge and Hjellbrekke, 2010; Fagerli and Hjellbrekke, 2008; Fagerli and Aas, 2007; 2008; Colette et al., 2020). In some measurement networks, the measurements are made periodically. For example, a lot of the US $PM_{2.5}$ speciation measurements are made once every 6 days. In these instances, we took the approach that the average of all such fixed frequency measurements in a year (subject to the 75% data capture threshold already described) represents an unbiased estimate of the annual average. This is, we believe, how the measurement network derives an annual average from such measurements when an annual average measurement value is provided.

It is not practical in a study such as ours that is considering multiple species from multiple networks to compare model and observations matched specifically to the native resolution of the measurements. The temporal resolution and frequency of the measurement data not only vary between different regions, but also for different species in a given network and sometimes also even for the same species at different sites in the same network. Aside from the enormous effort to undertake data matching, it would also mean that model-observation statistics are not on a comparable basis between networks and species. The following provides some illustrations of these issues. In the daily $PM_{2.5}$ speciation file reported by USEPA Air Data in 2010, $NH_4^+$ measurements were conducted every 3 days for 10 months of the year at the North Birmingham site, but at more-than-3-day intervals during January and December of that year. On the other hand, $NH_4^+$ measurements at the Wylam site were made at 6

day intervals. In addition, whilst the interval measurements at the North Birmingham and Wylam sites started on $2^{nd}$ January 2010, they started on $5^{th}$ January at the 'Alsup Elementry School - Commerce City' site. These inter-site differences mean that different coding script would be needed for data matching even for one species in one network. Moreover, North Birmingham (Latitude: 33.553056; Longitude: -86.815) and Wylam (Latitude: 33.499722; Longitude: -86.924167) are located in the same model grid ($1° \times 1°$ resolution), leading to another issue of how to compute a measurement average from multiple sites located in the same model grid that have different temporal resolution and/or measurement frequency. Measurements from the EMEP/CCC network also illustrate the variability in native resolution present in one network for one species. Again, taking $NH_4^+$ as an example, its measurement varies from hourly, daily, and 3-day to weekly, biweekly, and monthly among different sites. In addition to different temporal resolution, measurement sites in the EMEP/CCC network also differ in the starting date of the sampling period. For example, the starting date (in 2010) was at day = 0.125 at the Amberd site (daily measurements), at day = 1.375 at the Montseny site (daily measurements), and at day = -1.58 (from Day -1.58 to Day 12.42) at the Payerne site (biweekly measurements). The time interval between each sampling period at the same measurement location is also different at some sites. Taking Montseny site as an instance, the first 24-hour sampling during 2010 was from $1.375^{th}$ day to $2.375^{th}$ day, the second sampling was from $5.375^{th}$ day to $6.375^{th}$ day, while the third sampling was from $12.375^{th}$ day to $13.375^{th}$ day. Other differences arise through different data formatting protocols between the networks or even within the same network: for example, to represent an invalid measurement the number 99.99 is used for some EMEP sites (e.g., Illmitz, Melpitz, Tange), while the number 9.999 is used at other EMEP sites (e.g., Amberd, Ähtäri II, Yarner Wood). Or differences in date format between networks or in how time during the year is expressed: EMEP uses fractions of the year (at hourly discretisation, e.g., day 13.375), while hour-day-month date format is used by EPA Air Data.

All these differences make the coding of model-measurement time-matching not only close to unique for an individual site, but often unique to a given species at that site, and therefore cannot be readily automated. For multiple species in hundreds of measurement locations, the time and resource demand is not justified; and, as already noted above, would mean that model-observation data would not be directly comparable across sites, networks and species. This is why we take the pragmatic approach of averaging available data (when these are collected at a systematic measurement interval throughout the year) as the method to derive the value for the measurement annual (or monthly) average. We had previously drawn attention to the factual existence of variation in measurement averaging and periodicity in the Methods section (lines 160-168) and Discussion section (lines 704-713), but in view of the importance of this issue we have now added a couple more sentences to bolster this point in the Discussion (lines 714-716), as follows.

"In addition, different networks, and even different sites in the same network, may measure at different frequencies and at different times, which presents inherent practical difficulties in comparing model

simulations with ambient measurements. Further moves towards global standardised approaches for measurements across different networks is encouraged."

2. Given that one of the key uses of ACTMs such as WRF/EMEP-MSC-W is to quantify the effects of emission scenarios as stated by the authors in both the abstract and summary, an analysis of how well the model captured historic changes in concentrations and deposition (due to changes and variations in both emissions and meteorology) can help establish its credibility for that purpose. The two simulations performed for 2010 and 2015 with the ECLIPSEe emission inventories seem to offer such an opportunity. I strongly encourage the authors to consider adding a section that quantifies emission changes between 2010 and 2015 and then also compares observed and modeled concentration and deposition changes between the two years. While for some pollutants and regions the changes may be dominated by interannual meteorological variability, for others the concentration changes may show a clear link to emission changes.

Response: We thank the reviewer for making this useful suggestion. We have undertaken an evaluation of how the modelled concentrations and depositions respond to the change in emissions for the two simulations for 2010 and 2015 that use their respective ECLIPSE$_E$ emissions data. The results of this evaluation are reported as a new section of text, 4 figures and a table in the Supplementary Material (Pages 8-16) ("Evaluation of model response to changes between 2010 and 2015 ECLIPSE$_E$ emissions"). These comparisons of model and measurement responses between the two years provide useful additional confirmation that the model is behaving in line with expectations, subject to the following two caveats. First, the need for there to be measurement sites operating in both 2010 and 2015 substantially reduces the number of comparisons between the two years for some measurement networks and some species. Secondly, the comparison of the model (and measurement) changes between the two years with the emissions changes between the two years is confounded by any change in relevant meteorology between the two years (as the Reviewer acknowledges in their comment) and, for the secondary species, by the non-linear response of the chemical reactions to emissions changes, and the transport of species away from locations of emissions.

In our revised paper we draw attention to this additional model evaluation presented in the Supplementary Material via inclusion of a new Results subsection 3.1.3 entitled 'Evaluation of model response to changes between 2010 and 2015 ECLIPSE$_E$ emissions'. This subsection has the following text (lines 343-369).
"An evaluation was also undertaken of how the modelled concentrations and depositions respond to the change in emissions in simulations using the 2010 and 2015 ECLIPSE$_E$ emissions data, and of how these responses compared with the changes observed in the measurements between the two years. This analysis is presented and discussed in the Supplementary Material. Figures S6-S9 respectively present

global maps of the differences between 2010 and 2015 of the $NH_3$, $NO_x$ and $SO_x$ precursor emissions, the modelled $NH_3$, $NO_2$ and $SO_2$ concentrations, the modelled $NH_4^+$, $NO_3^-$ and $SO_4^{2-}$ aerosol concentrations, and the modelled total depositions of reduced N, oxidised N and oxidised S. Table S1 quantifies the trends between 2010 and 2015 in the modelled and measured species concentrations for each regional network for those sites where measurement data are available in both 2010 and 2015. The need in this comparison for measurement sites operating in both 2010 and 2015 severely reduces the number of paired comparison data for some measurement networks.

In summary, changes in emissions of $NH_3$ between 2010 and 2015 $ECLIPSE_E$ inventories are generally small (Figure S6). The global area-weighted average $NH_3$ emission increases by 4.5% from 2010 to 2015. By contrast, $NO_x$ and $SO_x$ emissions show slightly larger variations (Figure S6). The global area-weighted average emissions of $NO_x$ and $SO_x$ decrease from 2010 to 2015 by 5.7% and 14% respectively. The trends in modelled $NH_3$, $NO_2$, and $SO_2$ annual concentration changes between 2010 and 2015 (Figure S7 and Table S1) are entirely consistent with the trends in the emissions supplied to the model, and in the corresponding measurements, given both the realistic uncertainties in emissions and measurements (and the small number of measurement data), and the differential influences of meteorology on concentrations between the two years. Most parts of the world show increased $NH_3$ concentrations but decreased $NO_2$ and $SO_2$ concentrations from 2010 to 2015. The impacts of emission changes on modelled concentrations of secondary pollutants (Figure S8 and Table S1), and modelled total deposition of reduced N, oxidised N, and oxidised S (Figure S9) are varying. The comparison of modelled and measured concentration changes based on measurement locations (Table S1) indicates that trends in modelled and measured concentrations for $SO_2$ and $SO_4^{2-}$ in most networks from 2010 to 2015 show clear decreases, while for $NH_3$, $NH_4^+$, $NO_2$, $HNO_3$, and $NO_3^-$ the modelled and measured concentrations reveal a mixture of upward, downward and no trends but are again generally consistent with each other.

Overall, these comparisons of changes in model-simulated concentration and deposition between the two years in relation to the changes in measurements (and the emissions) provide useful additional confirmation that the model is behaving in line with expectations, within realistic levels of measurement uncertainty.

**Specific comments:**
Abstract, line 27: change "measurement" to "measurements"

Response: Requested change made.

Abstract, line 33: suggest changing "most" and "least" to "largest" and "smallest"

Response: Requested change made.

Section 1, line 49: Do we know which components (SIA, OM) drive this increase in PM2.5 burden reported by Shaddick et al. (2020)?

Response: Unfortunately, we do not find a statement in this paper regarding which components drive this increased $PM_{2.5}$ burden. The paper only provides estimates of $PM_{2.5}$ concentration data.

Section 1, lines 70 – 75: The work by Tan et al. (https://acp.copernicus.org/articles/18/6847/2018/) could also be referenced here

Response: We thank the reviewer for this suggestion, which we now include.

Section 2.1, line 87: consider changing "with implementation" to "used for applications"

Response: Requested change made.

Section 2.1, lines 104 – 112: Please provide additional information on the WRF configuration options and input datasets used in these simulations (e.g. land use / land cover database, microphysics scheme, cumulus parameterization scheme, PBL scheme, radiation scheme, land-surface model, etc.). Please also discuss whether the land use / land cover information used in the WRF simulations is consistent with the information used in the EMEP MSC-W biogenic emission and deposition calculations.

Response: The following information on the WRF configuration options and input datasets is now added to lines 107- 112 of revised manuscript.
"This work uses the Yonsei University (YSU) planetary boundary layer (PBL) scheme. The bulk microphysical parameterization (BMP) scheme is from Lin et at. (2011). The cumulus parameterisation uses the Kain-Fritsch scheme. The longwave and shortwave radiation scheme utilises RRTM/Dudhia. The WRF simulations used the Noah Land-Surface Model, and for land-cover setup, WRF uses the MODIS derived land cover and the EMEP MSC-W model uses land data from GLC2000 with the Community Land Model (CLM). The EMEP MSC-W model calculates roughness length and depositions from its own land cover."

Section 2.1, lines 133 - 149: It appears that no day-of-week or hour-of-day temporal profiles were applied to either the ECLIPSEe or HTAP emissions. If this is correct, what was the rationale for not applying such profiles?

Response: We now see that our description in Section 2.1 did not provide information specifically on this point. The EMEP MSC-W model default hour-of-day temporal profiles (which varies with SNAP sector) were applied to all countries. The EMEP MSC-W model default day-of-week temporal profile was applied to all of Europe but not elsewhere. Neither of the emissions inventories supplies such temporal information (far less on a regional or country basis) and as many countries do not use the same weekend system as Europe it would not be appropriate to apply European day-of-week profiles. In this study, we are focused on annual- and monthly-average concentrations and deposition which are relatively unaffected by the exact detail of the daily variations in emissions. On behalf of atmospheric modellers everywhere we support the development of country, source, and species temporal emissions profiles.

We have now added the following explicit statement on these aspects of emissions temporal profiles to the revised manuscript (lines 151-153).

"The EMEP MSC-W model default hour-of-day temporal profiles (which varies with SNAP sector) were applied to all countries. The default day-of-week temporal profile was applied to Europe only as neither of the emission inventories supplies such temporal information."

Section 2.2, lines 173 – 195: please see my first main comment regarding the use observations with their native temporal resolution.

Response: Please refer to our responses to the first main comment.

Section 2.2, lines 181 – 185: What was the rationale for not including wet deposition measurements from CAPMoN (https://www.canada.ca/en/environment-climate-change/services/air-pollution/monitoring-networks-data/canadian-air-precipitation.html ) to strengthen the analysis in Section 3.3?

Response: We thank the reviewer for drawing our attention to the CAPMoN wet deposition measurement network in Canada. We have added this to our network summary table (Table 1) in the Methods Section 2.2 together with some brief accompanying text description in this section (lines 191-194). Data from this network are now added to our model-observation comparisons reported in Section 3.3. This has resulted in additional model-measurement scatter plots of precipitation concentration and wet deposition comparisons in Figure 12 and Supplementary Figure S11, and corresponding additional model-measurement comparison statistics in Table 6 and Table S2. The correlation between model and measurements for this additional network are excellent, with $R$ values of 0.82 and 0.81 respectively for reduced N wet deposition and oxidised N wet deposition. The data in Table 7 in the Discussion section, which provides comparison statistics for all networks as a single dataset, have also been updated to

include this new network. Likewise for values in the text quoted from this table and for instances where we refer to the total number of measurement networks used in this study.

Section 2.2, lines 186 – 187: Please make sure to clearly distinguish between "R" and "RT" in the supplement and their first uses in the text, tables, and figures.

Response: We thank the reviewer for pointing out this potential ambiguity in referring to different calculations of a correlation coefficient. An explanation of the '$R$' and '$R_T$' notation has been added to the start of Section 3.2 (lines 374 – 378). The usage of $R$ with a subscript $x$ ($R_x$), refers to a particular network X, whilst the subscript $T$ ($R_T$), refers to a total dataset of more than one network. All usages of '$R_x$' and '$R_T$' in text, tables, and figures are updated accordingly. This notation is only used in Section 3.2, which we have also noted in the explanatory text at the start of this section.

Section 3.1.1., line 216: consider changing "total gridded differences" to "the total number of grid cells"

Response: Requested change made.

Section 3.1.2, lines 292 – 295: I suggest extracting and comparing the HTAP and ECLIPSEe emissions for the grid cells corresponding to these locations (maybe in addition to some surrounding grid cells) to gain further insights into the relative importance of potential localized emission differences (i.e. localized model uncertainty) vs. localized phenomena affecting the measurements.

Response: Whilst we can extract the emissions for these grid cells it is not possible to draw evidenced conclusions for their differences. We do not compile the emissions inventories and do not know the methods and underlying data used to estimate the emissions of a particular species in a particular spatial grid. The differences in emissions between the two inventories may arise because of different information on distributions of source types used by one inventory compared with another inventory (including the age of the underlying data). The difference may also be an unfortunate error introduced somewhere in the emissions compilation or its publication. We cannot know. Using satellite images to inspect the model grid might suggest some clues in regard to, for example, a point source of emissions in that grid that might be in one inventory but not in another, but this still wouldn't tell us what actually underpins the difference in the emission for that grid in one inventory versus another inventory.

Section 3.1.2, line 328: I suggest avoiding the term "bias" since the analysis presented here does not constitute a rigorous verification of emission inventories.

Response: 'Bias' is changed to 'difference' accordingly.

Section 3.2.1, lines 374 – 375: "there is no significant difference between model and measurements in most Southeast Asia countries" – This seems like an overstatement given that FAC2 is only about 0.5 for EANET NH3.

Response: We accept this comment, and this sentence is now deleted.

Section 3.2.1, lines 394 – 397: This again seems like an overstatement and selective reading of evaluation metrics. The FAC2 values for many of the EANET pollutant measurements are quite low, and many of these FAC2 values as well as NME and NMB values are worse for EANET than NNDMN.

Response: Our conclusion here was mainly based on the correlation coefficient. As we discuss in Section 4, there are good reasons why modelled and measured values differ, in particular at the spatial resolution inherent in global atmospheric chemistry models. Consequently, for this evaluation we place greater emphasis on the capability of modelling spatial trends as captured by the spatial correlation coefficient. A model-measurement comparison that has a large FAC2 value and small NMB and NME values, but poor correlation coefficient, may reflect closeness of the value in a less meaningful way. An atmospheric chemistry model's ability to produce the exact absolute value as a measurement is also driven to a large extent by two factors outside the model's control and hence beyond the scope of any model-measurement evaluation: first the accuracy of the emission inventory (including its spatial and temporal distribution); and, secondly, the accuracy of the measurement. From these perspectives, therefore, we feel that the correlation coefficient is more important than FAC2, NME and NMB for our model evaluation. In this particular section of the paper, the model shows better correlations with 2015 annual average measurements for EANET network (Mean R for 7 species: $\overline{R_7}$ = 0.73), than for the NNDMN network ($\overline{R_5}$ = 0.35). We have modified the text to make it more explicit to which statistical variable we are referring. It now reads as follows (lines 435-438).
"Linear correlations of aerosol components between model and EANET measurements are high ($R_E$ = 0.73-0.74). In summary, the model shows good performance in capturing spatial variations of key inorganic pollutants at EANET locations. The comparison statistics also show an overall better model-measurement linear correlation for EANET than for NNDMN for all species."

Section 3.2.3, lines 471 – 472: What was the rationale for not including AIRBASE over Europe? If it was included, the number and type of sites might be more comparable to the NO2 and SO2 measurement sites over North America.

Response: AIRBASE is dominated by urban sites that are close to emission sources. The model grids in the global model are of the order of 100 km in dimension and hence the AIRBASE sites will highly

likely not provide good representation of the model grid average. In fairness to AIRBASE sites, their purpose is not for evaluation of global models.

Section 3.2.3, lines 474 – 476: It is likely the location of the sites which are often source-oriented. SO2 emissions from the largest sector (power generation) are directly measured and included in the EPA emission inventory that was used in HTAP. Does the ECLIPSEe inventory not use this available information about measured SO2 emissions from the U.S. power sector?

Response: Yes, we assume that the emissions from these big point sources are directly included in the ECLIPSE emissions inventory as the Reviewer reports is the case for the HTAP inventory. We use the emissions inventories as provided, since the inventory developers are the professionals in this task. As we noted in an earlier response, differences in emissions between inventories may arise because of different information on distributions of source types used (or differences in emissions factors), or simply because of some error introduced somewhere.

Where there are larger differences between model and measurements at one site compared with other sites, this may be for a variety of reasons that we discuss above and in Section 4 of the paper, including that the measurement site is intentionally (or unintentionally) located so as to be influenced by a major local source or that the measurement happens to be in error. It is for these sorts of reasons that conclusions from model-measurement comparisons must be based on statistics that include as many sites as possible, and across a wide a geographical domain as possible. The presence of persistent model-measurement difference at a particular site for a particular species can act as a flag that emissions and/or site location and/or measurement accuracy needs investigation for that particular location.

Section 3.2.3, lines 480 – 481: As for SO2, this is likely due to the local-scale nature of many NO2 measurement sites.

Response: We agree, see the comment above.

Section 3.2.3, line 482: I suggest replacing "secondary" with "aerosol" since there are other secondary pollutants (e.g. HNO3) in Table 4 besides sulfate, nitrate, and ammonium.

Response: Suggested change made.

Section 3.2.4, lines 505 – 506: As noted in my first main comment, higher temporal resolution data is also available over the U.S. and Canada and should be used here as well.

Response: Please refer to our responses to the first main comment.

Section 3.2.4, Figure 11: Please add results for available North American networks (e.g. AMoN) here and in Figure S6.

Response: The measurement networks shown in Fig. 11 and Fig. S6 all have readily available monthly data that can be used for monthly comparison directly. We provide this monthly comparison for four networks. As discussed in our responses to the first main comment, producing monthly data based on measurements' native temporal resolution is an impractically time-consuming task, particularly when the temporal resolution and/or sampling frequency differ even within the same network. We would encourage the network organisations to provide their best estimate for monthly averages from their data (where possible) based on their best knowledge of the performance of their measurements. A useful monthly comparison of model and measurement data is also dependent on an accurate monthly temporal profile for the emissions. The emission inventories do not provide this information for many regions of the world.

Section 4, lines 666 – 676: Please see my first main comment about using observational data at their native temporal resolution for the analysis presented in this manuscript.

Response: Please refer to our responses to the first main comment.

Section 4, lines 704 – 705: "unmatched temporal coverage" should not be an issue if model-observation matching is performed at the native temporal resolution of the observations as discussed in my main comment.

Response: We accept that our use of "unmatched temporal coverage" is not fully correct here since, at least in principle, temporal matching is possible. However, please see our responses to the first main comment for detailed discussion of the problems doing this in practice for model evaluations comprising multiple species, measured at multiple sites, in multiple networks. We have now modified this phrase to "inconsistent temporal coverage" (line 743-744).

Section 5, lines 763 – 764: Please see my second main comment about adding an analysis of observed and modeled 2010 vs. 2015 changes in concentrations to evaluate the modeling system's ability to capture such changes which are at least partly driven by changes in emissions.

Response: We have now undertaken this analysis. Please refer to our responses on this to the second main comment.

**References cited in these responses**

Berge, H., and Hjellbrekke, A.-G.: Acidifying and eutrophying components: validation and combined maps. Supplementary material to emep status report 1/2010, available online at www.emep.int, The Norwegian Meteorological Institute, Oslo, Norway, 1-3, 2010.

Colette, A., Solberg, S., Aas, W., and Walker, S.-E.: ETC/ATNI Report 08/2020: Understanding Air Quality Trends in Europe. Focus on the relative contribution of changes in emission of activity sectors, natural fraction and meteorological variability, European Environment Information and Observation Network, 4-9, 2020.

Fagerli, H., and Aas, W.: Validation of nitrogen compounds in the emep model. In Transboundary acidification, eutrophication and ground level ozo ne in Europe in 2005. EMEP Status Report 1/2007, The Norwegian Meteorological Institute, Oslo, Norway, 73-90, 2007.

Fagerli, H., and Aas, W.: Using the EMEP intensive measurement data to evaluate the performance of the EMEP model for nitrogen compounds. In In Transboundary Acidification, Eutrophication and Ground Level Ozone in Europe in 2006. EMEP Status Report 1/2008., The Norwegian Meteorological Institute, Oslo, Norway, 109-126, 2008.

Fagerli, H., and Hjellbrekke, A.-G.: Acidification and eutrophication. In Transboundary Acidification, Eutrophication and Ground Level Ozone in Europe in 2006. EMEP Status Report 1/2008, The Norwegian Meteorological Institute, Oslo, Norway, 41-56, 2008.

Fagerli, H., Gauss, M., Benedictow, A. C., Steensen, B. M., and Hjellbrekke, A.-G.: Acidifying and eutrophying components: validation and combined maps. Supplementary material to emep status report 1/2011, available online at www.emep.int, The Norwegian Meteorological Institute, Oslo, Norway, 1-3, 2011.

Fagerli, H., Tsyro, S., Jonson, J. E., Nyíri, Á., Gauss, M., Simpson, D., Wind, P., Benetictow, A., Klein, H., Mortier, A., Aas, W., Hjellbrekke, A.-G., Solberg, S., Platt, S. M., Yttri, K. E., Tørseth, K., Gaisbauer, S., Mareckova, K., Matthews, B., Schindlbacher, S., Sosa, C., Tista, M., Ullrich, B., Wankmüller, R., Scheuschner, T., Bergström, R., Johanson, L., Jalkanen, J.-P., Metzger, S., van der Gon, H. A. C. D., Kuenen, J. J. P., Visschedijk, A. J. H., Barregård, L., Molnár, P., and Stockfelt, L.: Updates to the EMEP MSC-W model, 2018-2019, in: Transboundary particulate matter, photo-oxidants, acidifying and eutrophying components. EMEP Status Report 1/2019, The Norwegian Meteorological Institute, Oslo, Norway15046109 (ISSN), 32, 2019.

Lin, Y., and Colle, B. A.: A New Bulk Microphysical Scheme That Includes Riming Intensity and Temperature-Dependent Ice Characteristics, Monthly Weather Review, 139, 1013-1035, 10.1175/2010MWR3293.1, 2011.